# Dysregulation of Epigenetic Mechanisms of Gene Expression in the Pathologies of Hyperhomocysteinemia

**DOI:** 10.3390/ijms20133140

**Published:** 2019-06-27

**Authors:** Joanna Perła-Kaján, Hieronim Jakubowski

**Affiliations:** 1Department of Biochemistry and Biotechnology, Poznań University of Life Sciences, 60-637 Poznań, Poland; 2Department of Microbiology, Biochemistry and Molecular Genetics, Rutgers-New Jersey Medical School, International Center for Public Health, Newark, NJ 07103, USA

**Keywords:** hyperhomocysteinemia, DNA methylation, histone, homocysteine thiolactone, *N*-homocysteinylation, miRNA, epigenetic, atherosclerosis, Alzheimer’s disease, gene expression

## Abstract

Hyperhomocysteinemia (HHcy) exerts a wide range of biological effects and is associated with a number of diseases, including cardiovascular disease, dementia, neural tube defects, and cancer. Although mechanisms of HHcy toxicity are not fully uncovered, there has been a significant progress in their understanding. The picture emerging from the studies of homocysteine (Hcy) metabolism and pathophysiology is a complex one, as Hcy and its metabolites affect biomolecules and processes in a tissue- and sex-specific manner. Because of their connection to one carbon metabolism and editing mechanisms in protein biosynthesis, Hcy and its metabolites impair epigenetic control of gene expression mediated by DNA methylation, histone modifications, and non-coding RNA, which underlies the pathology of human disease. In this review we summarize the recent evidence showing that epigenetic dysregulation of gene expression, mediated by changes in DNA methylation and histone *N*-homocysteinylation, is a pathogenic consequence of HHcy in many human diseases. These findings provide new insights into the mechanisms of human disease induced by Hcy and its metabolites, and suggest therapeutic targets for the prevention and/or treatment.

## 1. Introduction

Hyperhomocysteinemia (HHcy), a medical condition of elevated homocysteine (Hcy) concentration in the plasma, usually defined as >15 µM [1,2], is prevalent in the general population and may have severe implications for the human health [3]. Cardiovascular disease and stroke [4], dementia [5], schizophrenia [6], infertility, pregnancy complications, birth defects [7], cancer [8,9], liver injury [10], and osteoporosis [11] are all associated with HHcy. Elevated Hcy is often present in patients with kidney disease and uraemia [12] and nonalcoholic fatty liver disease [13,14]. HHcy is a risk factor for atherosclerosis, a chronic inflammatory disorder of large and intermediate arteries. Hcy levels tend to increase with age and disease, including atherosclerosis and thrombosis [15]. HHcy causes vascular injury, manifested by endothelial desquamation, which in turn provokes smooth muscle cell proliferation leading to atherosclerosis as shown in a baboon model [16]. Two recent reviews provide an overview of pathologies associated with HHcy [17,18].

HHcy is a consequence of insufficient intake of B-vitamins (B_12_, B_9_, B_6_) and betaine (cofactors of Hcy-metabolizing enzymes), excessive intake of Met, the only known source of Hcy in the human body, as well as of mutations in genes encoding cystathionine β-synthase (CBS), 5,10-methylenetetrahydrofolate reductase (MTHFR), or methionine synthase (MS) [3]. Although several molecular mechanisms that can explain the toxicity of HHcy have been elucidated [2,9,19], it is still not entirely clear if there is causality between HHcy and associated diseases. The studies of molecular mechanisms of HHcy toxicity are complicated by the fact that there are multiple molecular forms of Hcy [20,21], each of which may exert different and sex-specific effects on biological molecules [22], tissues and organs [23,24]. 

HHcy has a wide range of biological effects and the pathophysiological changes associated with HHcy are most likely accumulated outcomes of Hcy and its metabolites, such as Hcy-thiolactone (HTL), *N*-Hcy-proteins, *S*-Hcy-proteins, S-adenosylhomocysteine (SAH), and low molecular weight disulfides of Hcy [23]. 

In the majority of Hcy-lowering clinical trials, so called ‘total Hcy’ (tHcy) was examined as a marker for outcomes. However, because tHcy is a complex marker, which nevertheless does not include other important Hcy metabolites (such as HTL, *N*-Hcy-protein, SAH), the conclusions from those trials may not be informative. A case in point is a recent study with a WENBIT cohort that found that HTL is a predictor of acute myocardial infarction but is not affected by tHcy-lowering therapy [25]. 

In the human body Hcy arises from dietary Met as a result of multiple reactions known as Met cycle (Figure 1). First, Met is activated to S-adenosylmethionine (SAM) by Met S-adenosyl transferase. SAM is a donor of methyl group to multiple acceptor molecules, e.g. proteins, DNA, RNA, lipids, and neurotransmitters, and after donating the methyl group, gives rise to SAH. SAH is subsequently hydrolyzed by SAH hydrolase (AHCY) to yield Hcy. Hcy is then metabolized via four reactions (Figure 1): remethylation to Met by Met synthase (MS) or betaine:Hcy methyltransferase (BHMT), transsulfuration to Cys by CBS and cystathionine γ-lyase (CSE), conversion to HTL by Met-tRNA synthetase (MARS), or oxidation with thiol groups to form *S*-Hcy-protein or low molecular weight disulfides [19]. Hcy, by being a byproduct of cellular methylation reactions, is a sensitive marker of one-carbon metabolism important for multiple physiological processes, including epigenetic regulation [26,27].

Epigenetic regulation, as a response to endogenous and exogenous factors, leads to alterations in gene expression via DNA methylation, histone modification, and the action of noncoding RNA. Epigenetic remodeling plays an important role in both normal and pathological conditions of an organism. Modifications of DNA and histones regulate gene expression by changing chromatin structure, making it transcriptionally permissive (euchromatin) or inactive (heterochromatin). 

Hcy occupies an important junction between one carbon metabolism and epigenetic processes (Figure 1). Methyl group used for methylation reactions originates form SAM, an intermediate in Hcy metabolism. Upon methylation SAM is converted to SAH, which inhibits transmethylation reactions. HHcy, through SAH accumulation and diminished methylation capacity (decreased SAM/SAH ratio) could lead to global hypomethylation [27]. However, this simplistic view is not supported by experimental evidence from cell culture (Table 1) and animal (Table 2) studies showing that decreased SAM/SAH ratio in general does not always lead to DNA hypomethylation [28,29].

A novel mechanism of epigenetic dysregulation in HHcy has been revealed by studies of Hcy-editing by MARS during protein biosynthesis [30], which generates HTL (Figure 1). HTL has a propensity to modify protein lysine residues [31,32], affording *N*-Hcy-protein (Figure 1), a process that also involves histones and interferes with their normal acetylation/methylation [19]. 

DNA and histone modifications as well as microRNAs and circular RNAs have been identified to have a crucial role in the progression of atherosclerosis [26,33], play an important role in stroke pathogenesis [34], Alzheimer’s disease (AD) development [35], neural tube defects [36], and cancer [37]. In this review we summarize the recent progress in studies of epigenetic dysregulation of gene expression by HHcy and its role in the etiology of human disease. 

## 2. HHcy Affects Gene Expression

One of the hallmarks of HHcy, observed in cell culture, experimental animal, and human studies, induced either by medium composition/diet or genetically, are profound changes in gene expression (Appendix A). The multiple pathways affected by Hcy and individual Hcy metabolites [38] are in agreement with the observations that HHcy is associated with multiple pathological conditions. In vitro cell culture experiments and in vivo animal models of HHcy show dysregulated expression of genes involved in certain pathological states, for example high expression of tau protein in hippocampus of high Met fed mice [39], or dysregulation of proteins linked to fatty liver disease [40,41], kidney disease [42,43], and AD [44,45]. 

How HHcy affects gene expression in experimental animals depends on many factors, including gender, type of diet, or mutations in genes involved in Hcy and related compounds metabolism. It is often found that some of the genes are differentially expressed as a result of both, genetic and dietary factors, while others are specific for a certain type of HHcy [40,41,42,43,44,45,46]. As an example, in mice AHCY increased in response to *CBS*^+/−^ genotype and/or diet (0.5% Met in drinking water), whereas glycine *N*-methyltransferase and BHMT only increased in response to diet. The top ranked network associated with the *CBS*^+/−^ genotype was the retinoid X receptor signaling pathway, while xenobiotic metabolism and the NRF2-mediated oxidative stress response were associated with the high Met diet [46].

The summary of changes in gene expression caused by HHcy or inactivation of genes involved in Hcy or HTL metabolism (Appendix A), shows that there is a broad range of processes affected. Upregulated pathways repeatedly occurring in in vitro and in vivo studies include basic metabolism processes, like glucose metabolism, amino acid and protein metabolism, oxidative stress response, and lipid metabolism. Downregulated pathways are also represented by a variety of processes, like lipoprotein, energy, and protein metabolism. 

As discussed in the following sections, the changes in gene expression induced by HHcy involve epigenetic mechanisms: DNA methylation and histone modifications, including acetylation, methylation, and *N*-homocysteinylation (Figure 1) (Table 1 and Table 2).

## 3. HHcy and DNA Methylation

In eukaryotes, DNA methylation occurs at the 5-position of cytosine residue (5mC) in CpG dinucleotide sites that frequently occur in the gene promoter region. DNA methylation is catalyzed by a family of DNA methyltransferases, which consists of DNA (cytosine-5)-methyltransferase 1, 3A, and 3B (DNMT1, DNMT3A, and DNMT3B, respectively). DNMT1 is responsible for maintaining methylation patterns established in the development and preferentially methylates hemimethylated DNA. DNMT1 uses the methylated DNA strand as a template to maintain DNA methylation during replication. DNMT3A and DNMT3B generate de novo DNA methylation (reviewed in [81]). DNMT3A interacts with DNMT1 and DNMT3B and is essential for the establishment of DNA methylation patterns during development and plays a role in paternal and maternal imprinting. DNMT3A has also other functions: binds to ZBTB18 transcriptional repressor [82], associates with HDAC1 through its ADD domain [83,84], interacts with histone H3 that is not methylated at K4 (H3K4) [85] and is recruited to trimethylated K36 of histone H3 (H3K36me3) sites [86]. The conformation of the active-site loop of DNMT3A is stabilized by DNMT3L, which also increases the binding of SAM [87].

Additionally, methylation of other regulatory elements, such as non-CpG regions, may also play a role in regulation of transcription, as reported for embryonic stem cells and adult tissues, for example in human skeletal muscle [88].

Another modification of DNA, namely hydroxymethylation of cytosine (5hmC), first identified in DNA of certain bacteriophages [89], rat, mouse and frog brain, rat liver and spleen [90], is present in genomes of many organisms [91]. In mammals, the highest levels of 5hmC are observed in the central nervous system, including brain mouse embryonic stem cells and postmitotic neurons [92,93]. Formation of 5hmC is catalyzed by Ten-Eleven Translocation (TET) family of methylcytosine dioxygenases via oxidation of 5mC [94]. Similarly to 5mC, 5hmC is an epigenetic marker and functions in regulation of gene expression [91]. Both genomic marks, 5hmC and 5mC, regulate transcription and their function may depend on DNA, chromatin and cell contexts [95].

DNA methylation depends on one carbon metabolism and is regulated by the availability of nutrients known as methyl donors. The effect of DNA methylation is often studied in animal models with altered content of methyl donors in the diet. The sensitivity of DNA methylation to dietary factors is sex- and organ-specific. Methyl donor-supplemented diet elevates folate intermediates in prefrontal cortex of 5-week-old mouse pups, whose mothers were fed high-fat or control diet through pregnancy and lactation. That increase, however, is blunted in male offspring from dams fed a high-fat diet. Pups of dams exposed to high-fat diet show increased concentration of Cys in the prefrontal cortex, suggesting an oxidative stress. Moreover, both maternal high-fat diet and postnatal methyl donor-supplemented diet increase global DNA methylation in the prefrontal cortex but only in males and not females. High-fat diet alone decreases methylation but this effect is also sex-specific [96]. 

### 3.1. SAM/SAH Homeostasis and DNA Methylation

DNA methylation is linked to Hcy metabolism through the generation of two metabolites: SAM, the donor of methyl group, and SAH, a byproduct of the methylation reaction. Although the SAM/SAH ratio is often considered an indicator of tissue methylation capacity, the relationship of SAM/SAH to gene-specific DNA methylation is tissue-specific and changes in DNA methylation can occur without concomitant changes in SAM/SAH levels [79,97]. Both dietary and genetic HHcy in animal models may, but not always leads to altered DNA methylation. 

### 3.2. HHcy & SAH/SAM Levels

HHcy may interfere with DNA methylation by affecting SAH/SAM levels. In animal models, HHcy may be associated with high SAH and low SAM/SAH ratio, but changes in SAM are not consistent in different models so SAM/SAH ratio is not a good proxy for DNA methylation levels in HHcy animal models [28]. Cbs-deficient (*Tg-I278T Cbs*^−/−^) mice with extreme HHcy throughout their early development have decreased SAM/SAH ratios in liver, heart, brain, and kidney but do not show signs of decreased global DNA methylation in those organs [28]. Similar situation is observed in an inducible mouse model of Cbs deficiency (*Tg-hCBS Cbs*^−/−^), in which severe HHcy can be induced temporally by withdrawing zinc from the drinking water. Although these mice exhibit elevated cellular SAH and reduced SAM/SAH ratios both in liver and kidney, this is not associated with alterations in the level of 5mC or various histone modifications (mono-, di-, trimethylations of H3K4, H4K9, H3K27, H3K36, and H3K79, acetylations H3K9ac, H3K14ac, H3K18ac, H3K56ac, and 2 phosphorylations H3Ser10p, H3Ser28p) in the liver [29].

Because of the structural similarity between SAH and SAM, SAH is a strong competitive inhibitor of DNA, RNA, and protein methyltransferases [98]. DNMT1, essential for maintenance of DNA methylation following DNA replication in cells, is strongly inhibited by SAH, generated during DNA methylation [99,100]. On the other hand, AHCY binds to DNMT1 during DNA replication and enhances DNMT1 activity in vitro. Overexpression of AHCY in mammalian cells leads to hypermethylation of the genome, whereas the inhibition of AHCY has an opposite effect. It has been suggested that alteration of AHCY level affects global DNA methylation and gene expression. Cells overexpressing AHCY exhibit up-regulation of metabolic pathway genes and down-regulation of PPAR and MAPK signaling pathways genes [101]. Human vascular endothelial cells upregulate AHCY as a response to HTL, Hcy, and *N*-Hcy-protein treatment [38]. Overexpression of human AHCY in the liver and kidney of HHcy mice lacking Cbs (*Tg-I278T Cbs*^−/−^) does not have significant influence on the phenotypes of *Tg-I278T Cbs*^−/−^ mice, and does not alter SAM/SAH homeostasis, meaning that the steady state concentrations of Met, tHcy, SAM, SAH, and SAM/SAH ratio in the liver and kidney have not been changed upon AHCY overexpression [102].

Feeding male Fisher 344 rats with a methyl-deficient diet (lacking of essential one-carbon nutrients, Met, choline, and folic acid) causes significant alterations of one-carbon metabolism, reflected in reduced concentrations of SAM, SAM/SAH ratios, accompanied by epigenetic changes—decreased global DNA methylation in the liver (Table 2). Whereas the changes in SAM, SAH levels are reversible, the DNA methylation can be normalized only after 9 weeks, but not after 18–36 weeks of the methyl-deficient diet, suggesting that long methyl deprivation leads to irreversible DNA hypomethylation [78].

In contrast to results obtained for the liver, in the brain of folate/methyl-deficient rats the levels of SAM, SAH, and SAM/SAH ratio do not differ from those of control animals. Only the concentration of Hcy is significantly higher after 36 weeks of folate/methyl deficiency. In spite of unchanged SAM/SAH ratio, the experimental diet causes global DNA hypermethylation in the brain reflected in elevation of genomic 5mC and methylation within unmethylated CpG-rich DNA domains. These alterations are concomitant with up-regulation of de novo DNA methyltransferase DNMT3A and methyl-CpG-binding protein 2. Gene expression profiling revealed that 36 weeks of folate/methyl-deficient diet caused differential expression of 33 genes in rat brains. The genes with changed expression are involved in inflammation and immune response, transcription, mitochondrial and carbohydrate metabolism, and nervous system development and function [79].

### 3.3. HHcy and DNA Methyltransferases

Hcy or any of its metabolites may also directly influence the expression and/or activity of DNA methyltransferases. However, also in this case the results are not entirely consistent. The direction of HHcy-induced changes in DNMT activity appear to depend on a studied model (Table 1 and Table 2). In human and mouse cell lines DNMT activity as well as expression is either stimulated or inhibited by HHcy. For example, HHcy due to CBS deficiency in human retinal endothelial cells (HRECs), and human retinal pigmented epithelial cells (ARPE-19), as well as in mice retina causes a significant, dose-dependent increase of DNMT activity [65]. Similarly, treatment with Hcy leads to up-regulation of DNMT1 protein in human umbilical vein endothelial cells (HUVECs) [49]. When mouse brain endothelial cells (bEnd.3) were treated with Hcy, increased expression of DNMT1 and DNMT3A was observed, while the expression of DNMT3B was decreased [67]. 

In contrast, other studies show opposite effects. For example, vascular smooth muscle cells (VSMC) treated with Hcy exhibit reduced expression and activity of DNMT1 [51]. HUVECs incubated with Hcy have decreased DNMT1 activity while DNMT3 activity is not significantly affected by Hcy [53]. Similarly, studies using animal models show inconsistent results (Table 2). In *ApoE*^−/−^ mice fed with a high-Met diet and in Tg-127T Cbs^−/−^ mice, HHcy decreases DNMT1 levels and reduces DNMT1, DNMT3A, and DNMT3B levels, respectively [57,70]. On the other hand, *Cbs*^+/−^ mice and NASH rats both having elevated plasma Hcy, have increased DNMT1 mRNA and protein [64,75], while F344 rats fed methyl-deficient diet exhibit decreased DNMT1, DNMT3A and DNMT3B proteins [79].

### 3.4. HHcy and DNA Methylation in Disease 

#### 3.4.1. Atherosclerosis

Atherosclerosis is a vascular inflammatory disease featured by narrowing of blood vessel lumen caused by the accumulation of lipid, fibrous materials, extracellular matrix protein and platelet; as well as inflammatory response, oxidative stress, endothelial cells (ECs) dysfunction, VSMC activation and proliferation; and shorter telomeres. Endothelium lining the blood vessel plays a key role in vascular function, e.g., mediates of blood vessel tone, hemostasis, neutrophil recruitment, hormone trafficking, and fluid filtration [103]. 

Many atherosclerosis hallmarks have been shown to be regulated in HHcy by epigenetic mechanisms. Starting with lipid metabolism, HHcy-induced atherosclerosis in *ApoE^-^*^/-^ mice is mediated by decreased DNA methylation and increased fatty acid-binding protein 4 (FABP4) expression (Table 2). FABP4 is a lipid transport protein in adipocytes and macrophages, that delivers long-chain fatty acids and retinoic acid to their cognate receptors in the nucleus and is involved in lipid metabolism and inflammatory response. High levels of FABP4, observed in atherosclerotic lesions, enhance the accumulation of cholesterols and triglycerides. *ApoE*^−/−^ mice fed an high-Met diet for 20 weeks had decreased expression of DNMT1 and increased serum SAH levels that may play a role in *FABP4* promoter hypomethylation. In fact, HHcy upregulates FABP4 in *ApoE^-^*^/-^ mice by inducing demethylation of the CpG sites of FABP4 promoter. Similar results were observed in cultured foam cells grown in the presence of Hcy (Table 1); *FABP4* promotor DNA methylation decreased significantly, while FABP4 mRNA and protein expression were up-regulated [57].

HHcy induces oxidative stress in HUVECs via changing the DNA methylation level. HUVECs treated with Hcy have decreased DNA total methylation level, which increases after treatment with folic acid and vitamin B_12_. On the other hand, the methylation level of SH3 domain-containing protein 1 (SORBS1) is higher in HHcy conditions and gets lower upon treatment with folic acid and vitamin B_12_. The HHcy-induced changes in DNA methylation are accompanied by increased malonyldialdehyde (MDA) level, up-regulated ICAM-1 expression, and reduced superoxide dismutase (SOD-2) and eNOS levels [47].

Oxidized low-density lipoprotein receptor 1 (LOX-1) is a vascular scavenger receptor, that mediates the recognition, internalization and degradation of oxidatively modified low density lipoprotein (oxLDL) by vascular ECs. By mediating proatherogenic responses, such as endothelial activation and dysfunction, smooth muscle cell proliferation and migration, phagocytosis of apoptotic cells, inflammation, cholesterol uptake and foam cell formation, collagen deposition, LOX-1 plays an important role in atherosclerosis development. The association of LOX-1 with oxLDL induces the activation of transcription factor NF-κB [104]. 

Hcy-induced oxidative stress occurs through TLR4/NF-κB/DNMT1-mediated LOX-1 DNA methylation in ECs. ECs treated with Hcy exhibit increased expression of toll-like receptor 4 (TLR4). TLR4 mediates the innate immune response against bacterial lipoproteins and other microbial cell wall components, which is important in the defense mechanism against microorganisms. Hcy downregulates DNMT1 expression, which induces LOX-1 DNA hypomethylation and elevates LOX-1 expression. Hcy also induces apoptosis and oxidative stress, manifested by increased levels of MDA, H_2_O_2_, and ox-LDL. Further, Hcy promotes the secretion of NF-κB, which by association with DNMT1, decreases DNA methylation [50].

Endothelial dysfunction is defined as decreased synthesis, release, and/or activity of endothelium-derived NO. During the development of atherosclerosis ECs produce chemoattractant factors that induce monocyte recruitment and infiltration in the neointima; monocytes differentiate into macrophages, that are charged with oxidized low-density lipoproteins (LDLs), leading to the formation of inflammatory foam cells; activated macrophage release inflammatory mediators; VSMC migrate and proliferate in intimal layer [103]. 

HHcy causes endothelial dysfunction and apoptosis, which play an important role in the development of atherosclerosis. One of the factors contributing to ECs apoptosis observed in atherosclerosis is increased level of the circulating asymmetric dimethylarginine (ADMA), an endothelial nitric oxide synthase (eNOS) inhibitor [105]. ADMA is degraded by dimethylarginine dimethylaminohydrolase (DDAH) [106]. There are two isoforms of DDAH with different tissue distribution. DDAH2 by decreasing ADMA level is believed to have protective effect on endothelial function [107]. 

HUVECs treated with Hcy exhibit increased level of ADMA and elevated apoptosis rate. Additionally, in Hcy-treated HUVECs protein level of DNMT1 is up-regulated, mRNA level of DDAH2 is down-regulated and *DDAH2* promoter methylation was increased significantly (Table 1) [49]. Notably, mildly elevated Hcy (10 and 30 μM) induces hypomethylation, while higher concentrations of Hcy (100 and 300 μM) cause hypermethylation in the promoter CpG island of *DDAH2* gene. Similarly, the effect on *DDAH2* expression depends on Hcy level, being increased in lower and decreased in higher Hcy concentrations, respectively. Other effects of Hcy, following the inhibition of DDAH2 activity, include the increase of ADMA concentration, the reduction of eNOS activity, and the decrease of NO production [48].

Elevated plasma SAH induces endothelial dysfunction via epigenetic up-regulation of the p66shc-mediated oxidative stress pathway. Mice with elevated level of SAH display impaired endothelium-dependent vascular relaxation and elevated production of reactive oxygen species (ROS) and p66shc expression in the aorta. Inhibition of AHCY induces hypomethylation in the *p66shc* gene promoter and inhibits the expression of DNMT 1 [108].

Endothelial dysfunction in HHcy can also be caused by reduced methylation of the *p66shc* gene promoter and increased p66shc expression mediated by inhibition of DNMT3B, as shown in HUVECs. Hcy-stimulated p66shc expression increases ROS, decreases nitric oxide, and induces up-regulation of endothelial intercellular adhesion molecule-1 [52]. 

HHcy is also proposed to cause endothelial dysfunction via a mechanism involving DNA hypomethylation and suppression of EC growth by transcriptional inhibition of the *cyclin A* gene. Treatment with Hcy inhibits *cyclin A* gene transcription and cell growth by inhibiting DNA methylation through suppression of DNMT1 in ECs [53]. The discrepancy between the inhibitory effect of Hcy on DNMT1 observed in [53] and the stimulatory effect in [49] with the same type of cells (ECs) can be explained by the inclusion of adenosine and an inhibitor of adenosine deaminase in [53], an artificial experimental setting, which has little to do with pathophysiology of HHcy. 

ECs dysfunction is linked with ECs activation, which is characterized by up-regulation of cell-surface leukocyte adhesion molecules, such as VCAM-1, ICAM-1, and endothelial leukocyte adhesion molecule (ELAM, also known as E-selectin). ECs activation is induced by proinflammatory cytokines, e.g., TNF-α, IL-6, and facilitates the recruitment and attachment of circulating leukocytes to the vessel wall [109]. Activated endothelium has proinflammatory and procoagulant features.

Treatment of HUVECs with Hcy (25–200 µM, 24 hours) promoted demethylation of soluble epoxide hydrolase (sEH) promoter and ATF6 binding to the promoter of *sEH*. sEH is a major enzyme that hydrolyzes epoxyeicosatrienoic acids and attenuates their cardiovascular protective effects. As a result, sEH mRNA and protein are elevated. Similar effect was observed in vivo in the aortic intima of mice with mild HHcy (induced in sEH^−/−^ mice with 2% (wt/wt) L-Met in a chow diet for 4 weeks), where expression of sEH and adhesion molecules are elevated. Additionally, Hcy-induces EC activation, evidenced by upregulated vascular cell adhesion molecule-1 (VCAM-1) and intercellular adhesion molecule-1 (ICAM-1), was markedly attenuated by sEH inhibition or gene deletion [54].

The development of atherosclerosis is characterized by VSMC proliferation. Hcy activates VSMCs by aberrant secretion of mitogen platelet-derived growth factors (PDGFs) from ECs in human and in mice. HHcy (100 µmol/L) caused proliferation and migration of VSMCs co-cultured with human ECs. Hcy upregulates mRNA levels of PDGF-A, -C and -D in ECs and reduces the expression and activity of DNMT 1 (Table 1), causes demethylation of PDGF-A, -C and -D promoters and enhances the binding activity of transcriptional factor SP-1 to the promoter. The up-regulation of PDGF by Hcy was confirmed in the aortic intima of mice with HHcy (C57BL/6J mice fed standard mouse chow diet with 2% L-Met for 4 or 8 weeks) (Table 2). Similar effect was observed in patients (n = 30) being at high risk of cardiovascular events humans, where HHcy was a predictor of increased serum PDGF level [51].

HHcy, being a risk factor for atherosclerosis, increases the proliferation rate of VSMC. Signaling networks controlling cell proliferation, survival and inflammation in VSMC are induced by phosphatase and tensin homologue on chromosome 10 (PTEN), which is endogenously expressed in VSMCs. Hcy, via up-regulation of DNMT1, increases the methylation rate of PTEN, and decreases its expression in VSMCs [59]. Atherosclerotic plaques in HHcy *ApoE*^−/−^ mice and Hcy-treated VSMCs have increased proportion of cells in S phase. Cells can be blocked in the G0/G1 stage of the cycle by transmembrane GTPase, mitofusin-2 (MFN2). VSMCs treated with Hcy decreases expression of MFN2, due to increased DNA methylation level of *MFN2* promoter. On the other hand, atherosclerotic plaques in HHcy *ApoE*^−/−^ mice and Hcy-treated VSMCs exhibit elevated expression of transcription factor c-Myc. In Hcy-treated VSMCs c-Myc binds to and up-regulates *DNMT1* promoter, which leads to *MFN2* promoter hypermethylation, and down-regulated MFN2 expression [62].

In cultured monocytes, treatment with Hcy elevates SAH, lowers SAM levels and the SAM/SAH ratio, and increases the activity of C-5MT-ase. Hcy treatment also increases the levels of peroxisome proliferator-activated receptor α and γ (PPARα,γ) promoter methylation and decreases mRNA and protein levels of PPARα,γ [58]. PPARα is a ligand-activated transcription factor and a key regulator of lipid metabolism [110]. PPARs are a group of nuclear receptor proteins that after activation by ligand (one of peroxisome proliferators) and dimerization with retinoid X receptor (RXR), regulate gene expression through binding to DNA of target genes. PPARγ has been implicated in the pathology of numerous diseases including obesity, diabetes, atherosclerosis, and cancer. PPARγ activates the *PON1* gene, increasing synthesis and release of PON1 from the liver and reducing atherosclerosis [111]. 

Bone marrow (BM)-derived endothelial progenitor cells (EPCs) play an important role in angiogenesis and vascular function and their function is regulated by CBS. BM-EPCs from mice fed with high Met diet exhibit hypermethylation of *CBS* promoter and the methylation level is negatively correlated with CBS mRNA and angiogenic function of BM-EPCs. High Met diet increases global methylation (5mC) and DNMT1 expression [66].

Leukocyte telomere length (LTL) is decreased in patients with atherosclerosis and negatively correlates with tHcy [72,112]. Clinical observations and HHcy mouse model studies indicate that Hcy induces DNA demethylation and down-regulation of TERT and further contributes to LTL shortening. In a mouse model HHcy is associated with mouse telomerase reverse transcriptase (*mTERT)* promoter demethylation and decreased mTERT mRNA expression [72]. LTL significantly decreases with the Hcy level in patients with hypertension but not in healthy subjects. Age, hypertension, tHcy and LDL combined contribute to LTL shortening. Patients with essential hypertension have lower methylation ratio of human telomerase reverse transcriptase (hTERT) and mRNA expression than healthy controls. HHcy promotes *TERT* DNA hypomethylation and reduces mRNA levels, which contributes to shortened LTL also in a hypertension rat model [73]. ECs subjected to chronic HHcy exhibit up-regulation of cellular senescence markers, p16, p21, and p53. Hcy affects also telomere length, promoting their shortening, by reducing *hTERT* promoter methylation and expression. Demethylation of *hTERT* promoter increases CCCTC-binding factor repression and interfering in the SP1 binding to the demethylated *hTERT* promoter. The mechanism of hTERT decreased methylation and expression related to CCCTC-binding factor have also been confirmed in a mouse model of HHcy [55].

The association of HHcy with hypertension and stroke [113] was found to depend on DNA methylation [114]. Specifically, a study examining the effects of methylation on the development of hypertension and stroke (stroke n = 132, hypertension n = 243, controls n = 218) has found that hypertensive and stroke patients had higher methylation levels in the *CBS* gene promoter compared with healthy controls. Additionally, the *CBS* promoter hypermethylation increased the risk of hypertension and stroke, especially in male patients [114].

#### 3.4.2. Uraemia

HHcy is prevalent in patients with uraemia, which may indicate aberrant methylation. In fact, in a study involving men with HHcy and uraemia treated with standard haemodialysis (n = 32), and healthy male controls (n = 11), total DNA hypomethylation was higher in patients than in controls and allelic expression was changed in both sex-linked and imprinted genes. The shift from monoallelic to biallelic expression was dependent on Hcy concentrations. Folate therapy, which reduces HHcy, restored DNA methylation to normal levels, and corrected the patterns of gene expression [115].

#### 3.4.3. Cognition and Alzheimer’s Disease

A review of studies on HHcy in relation to vascular contributions to cognitive impairment and dementia (VCID) has shown that Hcy at concentrations <100 μmol/L is associated with vessel wall fibrosis, myocyte proliferation, altered nitric oxide signaling, superoxide generation, and procoagulant actions [116]. The two most common types of dementia are Alzheimer’s disease (AD) and vascular dementia. Aberrant DNA methylation may be implicated in the AD pathogenesis. The development of AD could be associated with β-amyloid formation. A crucial factor for β-amyloid formation is the catalytic subunit of the γ-secretase complex, presenilin-1 (PS1), which catalyzes the intramembrane cleavage of integral membrane amyloid-beta precursor protein (APP). AD is associated with HHcy and altered SAM cycle. Demethylation of *PS1* promoter upregulates PS1 expression and γ-secretase activity, which in turn increases β-amyloid production. On the other hand, exogenous SAM, by restoring DNA methylation in *PS1* promoter, downregulates PS1 gene expression, and β-amyloid production [117].

In addition to PS1, the expression of BACE (β-secretase) is also regulated by DNA methylation. Neuroblastoma cells grown in culture medium with reduced levels of folate and vitamin B_12_ have altered Hcy metabolism as demonstrated by decreased intracellular SAM levels, increased PS1 and BACE levels, and β-amyloid production. This effect could be reversed by the administration of SAM to the deficient medium, which normalizes gene expression and reduces β-amyloid expression [118].

5-Lipoxygenase (5LO) catalyzes the first step in leukotriene biosynthesis, and thereby plays a role in inflammation [119]. 5LO is highly expressed in cerebral cortex and hippocampus, increases with age [120], and gets upregulated in AD brains [121]. Transgenic mice (3xTg-AD) overexpressing neuronal 5LO have impaired memory and develop AD-related tau pathology [122]. HHcy in mouse dietary and genetic models (male mice Tg-128T Cbs^−/−^) results in a significant increase of the SAH/SAM ratio and up-regulation of 5LO protein and mRNA. This is accompanied by reduced expression of DNA methyltransferases (DNMT1, DNMT3A, DNMT3B) and hypomethylation of *5LO* DNA. In vitro studies with N2A cells confirm these findings and demonstrate that the mechanism of the Hcy-dependent 5LO activation and β-amyloid formation involves DNA hypomethylation. This mechanism may underlie the pathogenesis of AD [70]. However, the conclusion that DNA hypomethylation is secondary to the elevated levels of SAH is not tenable because it can be explained by reduced DNMT expression, which is known to reduce DNA methylation.

Treatments of mouse brain endothelial cells (bEnd.3) with Hcy induces mitochondrial toxicity and endothelial dysfunction in part by epigenetic alterations. Exposure of cells to Hcy elevates mRNA and protein levels of *N*-methyl-d-aspartate receptor-1 (NMDA-R1), cellular Ca^2+^, NADPH-oxidase-4 (NOX-4) expression, mitochondrial dehydrogenase activity and decreases the level of nitrite, SOD-2 expression, mitochondria membrane potentials, and ATP production. Hcy also increases the expression of DNMT1, DNMT3A, but decreases DNMT3B expression, which may indicate increased DNA methylation. In addition, Hcy treatment induces the formation of autophagic vacuoles, upregulates LC3-I/II, and downregulates Atg3/7 and p62, which is consistent with increased mitophagy [67].

The data on global DNA methylation in AD patients are population- and brain region-dependent. Genome-wide DNA methylation in peripheral blood mononuclear cells (Caucasians living in Northern Italy, 37 subjects with late-onset AD and 44 healthy controls) has been found to be elevated in AD patients and associated with worse cognitive performances. AD patients exhibited upregulated DNMT1 mRNA and protein levels [123]. In a study with a cohort of elderly Polish individuals (n = 194), patients with dementia had significantly higher concentrations of Hcy and methylmalonic acid and lower folate and plasma 5-methyltetrahydrofolate than non-demented subjects. However, there was no difference in DNA methylation between patients and controls. Only a non-significant tendency to higher DNA methylation in patients with vascular dementia was observed [124].

Some studies have shown diminished overall DNA methylation and hydroxymethylation in brains of AD patients; in the hippocampus [125], entorhinal cortex and cerebellum [126], entorhinal cortex layer II [127]. Other researchers, however demonstrate opposite effects [128]. 5mC and 5hmC were elevated in the human middle frontal gyrus and middle temporal gyrus of brains from AD subjects in comparison with healthy controls and correlated with markers of AD including β-amyloid, tau, and ubiquitin loads. Further analyses of AD and control brains revealed that levels of 5mC and 5hmC were elevated in neurons and low in astrocytes and microglia [128].

Changes in DNA methylation are also observed in AD-related genes. For example, in AD patients and animal AD models some genes are hypermethylated (*MTHFR*, *APOE**ε4*, *MAPT*, *SORBS3*, *Tau*), while others are hypomethylated (*CREB5*, *S100A2*, *PP2A*, *BACE*, *PS1*) or exhibit no significant difference (*APP*) [35,129].

HHcy is a risk factor for dementia [130]. B vitamin supplementation by lowering Hcy protects against declining cognition, which is affected by genetic polymorphism (R278G, A/G) in the DNA methyltransferase gene *DNMT3L*. R278G affects the formation of DNMT3A-3L-H3 complex required for DNA methylation. Subjects involved in the VITACOG study, exhibiting mild cognitive impairment, who were supplemented for 2 years with vitamins B_6_, B_12_ and folic acid and carried the G allele of DNMT3L gene presented better “visuospatial associative memory” and slower rates of brain atrophy. On the other hand, among healthy, middle age individuals participating in the TwinsUK study, those who regularly took vitamins and were A/A homozygotes, had displayed improved “visuospatial associative memory”. Conclusion from those two studies it that in A/A homozygotes, while not having mild cognitive impairment, vitamin supplementation improves “visuospatial associative memory”, whilst in the older age when mild cognitive impairment begins, B vitamin treatment is associated with “visuospatial associative memory” decline. In G carriers, however, vitamin supplementation has no significant effect in the middle age, and improves “visuospatial associative memory” in the old age [131].

### 3.5. HHcy and DNA Methylation in Humans

The findings in animal and cell culture models that HHcy alters gene expression by affecting the methylation of specific genes, leads to a question whether HHcy alters global and site-specific DNA methylation in humans. However, the data on a relationship between genomic methylation and HHcy in humans are not consistent and seem to be population-dependent. Further, clinical trials show that Hcy-lowering does not normalize DNA methylation patterns [15]. 

The association between plasma tHcy and epigenome-wide DNA methylation was investigated in leukocytes of individuals from six cohorts (n=2,035). Three differentially methylated positions cg21607669 (SLC27A1), cg26382848 (AJUBA), and cg10701000 (KCNMA1) at chromosome 19, 14 and 10, respectively, were significantly associated with tHcy. In those three differentially methylated positions, increase in methylation was associated with elevated tHcy level. In addition, 68 Hcy-associated differentially methylated regions were identified, the most significant of which was a 1.8-kb domain at chromosome 6, containing 55 CpGs in gene *TNXB* and *ATF6B*. The sixty-eight differentially methylated regions were annotated to the 114 genes involved in 14 pathways (e.g. metabolic pathways, folate biosynthesis, glycosaminoglycan biosynthesis—heparan sulfate, phagosome and MAPK signaling pathway) [132]. The *TNXB* gene encodes an extracellular matrix glycoprotein (Tenascin XB) that may regulate the production and assembly of certain types of collagen and regulate the structure and stability of elastic fibers. TNXB deficiency is observed in the Ehlers-Danlos syndrome, a connective tissue disorder [133]. The *ATF6B* gene encodes cyclic AMP-dependent transcription factor ATF-6 beta that acts in the unfolded protein response (UPR) pathway by activating UPR target genes induced during ER stress. This gene is involved in the unfolded protein response during endoplasmic reticulum stress and has been shown to be activated by Hcy treatment in human ECs [54]. More specifically, a putative ATF6-binding motif is identified and shown to be demethylated upon treatment with Hcy. 

SLC27A1 (also known as long-chain fatty acid transport protein 1, FATP1), whose methylation is associated with HHcy, mediates the ATP-dependent import of long-chain fatty acids into the cell by mediating their translocation at the plasma membrane [134]. FATP1 binds to phosphorylated form of glutamyl-prolyl-tRNA synthetase (EPRS) inducing its translocation to the plasma membrane and long-chain fatty acid uptake. EPRS is phosphorylated by mammalian target of rapamycin complex 1 (mTORC1) and p70 ribosomal protein S6 kinase 1 (S6K1). Thus, EPRS and FATP1 are terminal mTORC1-S6K1 axis effectors that are critical for metabolic phenotypes and contribute to adiposity and aging [134]. Hcy may affect expression of genes coding those proteins via changes in DNA methylation and in this way influence connective tissue structure, endoplasmic reticulum stress, adiposity, and autophagy. 

The synthesis of Met from Hcy requires a methyl donor, 5-methyltetrahydrofolate (5-methylTHF), whose synthesis is catalyzed by MTHFR. The common c.677C>T polymorphism in the *MTHFR* gene, resulting in reduced enzymatic activity, under low folate plasma levels conditions, leads to higher tHcy levels. Studies of a cohort of northern Italian subjects homozygous for 677T (n = 105) and 677C (n = 187) *MTHFR* genotypes show that genomic DNA methylation in peripheral blood mononuclear cell correlats directly with the folate status and inversely with plasma tHcy concentrations. Subjects with TT genotypes and low folate levels have significantly reduced level of methylated DNA [135].

An association of site-specific changes in DNA methylation in humans and the interaction between mild HHcy and the c.677C>T polymorphism have been studied. A meta-analysis involving 2 cohorts (TwinsUK registry, n = 610 and Rotterdam study, n = 670) found 13 methylation probes significantly associated with *MTHFR* gene c.677C>T polymorphism × tHcy levels. These included sites on chromosome 1, 2, 3, 4, 7, 12, 16, and 19. The cluster of probes at the *AGTRAP–MTHFR–NPPA/B* gene locus on chromosome 1 had the most significant associations. Moreover, the top 2 hits on chromosome 1 were functionally associated with variability in expression of the *TNFRSF8* gene coding TNF receptor superfamily member 8. In the data of individuals from the TwinsUK registry there was a tendency for a negative relationship between genomic methylation and tHcy but only for subjects with the CT/TT genotype and not for those with the CC genotype. On the other hand, there was no relationship between genomic methylation and tHcy levels in the Rotterdam cohort [136]. A hypothesis that mild HHcy is associated with widespread methylation changes in leukocytes was not confirmed in a study of methylation levels in individuals of European ancestry from 12 cohorts (n = 9,894). There were no widespread changes in DNA methylation across the genome of analyzed subjects [137].

Vegetarians have often a low vitamin B_12_ intake, low content of Met in the diet, and exhibit elevated Hcy concentration which can induce the generation of SAH, an inhibitor of methyltransferases. In a study involving apparently healthy vegetarians (n = 71; lacto-/lactoovovegetarians n = 48 and vegans n = 23) a significant inverse correlation between SAH or the SAM/SAH ratio and whole-genome methylation was observed but no significant correlation was observed between Hcy and SAH, SAM or the SAM/SAH ratio. Also, the methylation status was not correlated with Hcy or SAM. In spite of an inhibitory effect of SAH on whole-genome methylation, no interaction between vegetarian lifestyle and DNA methylation could be found [138].

White matter hyperintensity (WMH) is a pathological change of the brain observed very often on magnetic resonance imaging and computed tomography in elderly and in subjects with stroke and dementia. WMH is often associated with cognitive impairment. It has been recently reported that patients with WMH (n = 140, and controls, n = 70) have significantly higher serum tHcy levels, and showed hypermethylation and reduced mRNA expression of *ERα-A* gene, as compared with the controls. In the analyzed group of people, the severity of WMH, hypermethylation of *ERα-A* gene and Hcy were independent risk factors of cognitive impairment. The methylation in *ERα-A* gene was positively correlated with Hcy concentration, the severity of WMH and cognitive impairment [139].

## 4. Histone Modifications in HHcy

### 4.1. Histone Modification

In addition to DNA modifications, posttranslational histone tail modifications have an impact on chromatin structure and affect gene expression. Histone modifications regulate DNA replication, transcription and repair by providing a platform for the recruitment of transcriptional regulators or chromatin remodellers [140]. Posttranslational histone modifications can serve as a repressive or activating signal, depending on the site on the histone tail being modified, number of groups added, and the region of chromatin where the modification occurs (for example, promoter versus intergenic regions). There are over a hundred known enzymes that modify 24 histone Lys and Arg residues [141] and at least eight different classes of histone modifications characterized, including acetylation of Lys, methylation of Lys (mono, di, tri), methylation of Arg (mono, di), phosphorylation of Ser, Thr and Tyr, ubiquitination and sumoylation of Lys, ADP ribosylation of Glu, deimination of Arg and Pro isomerization. Many different sites have been identified for each class. Functionally, histone modifications either disrupt chromatin contacts or affect the recruitment of other proteins to chromatin. Their presence dictates the higher-order chromatin structure, in which DNA is packaged and can orchestrate the ordered recruitment of enzyme complexes to process DNA. In this way, histone modifications have the potential to influence many fundamental biological processes, some of which may be epigenetically inherited [142]. 

Acetylation and methylation are the most studied histone modifications. Acetylation of histone tails is an important epigenetic regulation that adds negative charge to histones and is typically characteristic of transcriptionally active, non-compact chromatin [141]. Acetylation of nucleosomal histones affects chromatin organization and plays a key role in the switch between permissive and repressive chromatin structure [143]. The enzymes that add and remove the acetyl group from histone tails are called histone acetyltransferases (HATs) (31 known enzymes) and histone deacetylases (HDACs) (18 known enzymes), respectively [144]. Acetylation has been identified on 16 Lys residues (but not Arg residues) of *N*-terminal histones, including 3 Lys residues on H2A, 2 Lys residues on H2B, 7 Lys residues on H3, and 4 Lys residues on H4 [141]. 

Unlike most histone tail acetylations, histone tail methylations can be activating or repressing, depending on a specific position. As with other histone tail modifications, there are numerous enzymes involved in methylating or demethylating histone tails [81]. There are specific histone Lys methyltransferases (55 known enzymes) and demethylases (24 known enzymes) that act on mono-, di-, or trimethylation states. Methylation of histones occurs on Arg or Lys residues and alters binding of gene-expression regulating proteins due to enhanced hydrophobicity. The effect of histone Lys methylation depends on the position and the extent of the modification. Histone methylation occurs on 7 Lys residues and 5 Arg residues of *N*-terminal H3 (10 residues) and H4 (two residues) tails [141]. Methylation of H3K4 and H3K79 is commonly associated with gene activation while sites for gene inactivation include H3K9 and H3K27. Four lysine residues of histone 3 (H3K9, H3K23, H3K27, and H3K36) can be either methylated or acetylated. SAM is a common substrate for all methylation enzymes but the effect of SAM levels on histone modification can be site specific [144]. 

### 4.2. Histone-Modifying Enzymes in HHcy

Histone modification, an important modulator of gene expression, depends on the activity of histone-modifying enzymes. The expression of those enzymes is tightly regulated in a tissue-specific manner. A study of panoramic expression profile of 164 enzymes in 19 human and 17 murine tissues revealed that only certain tissues have high expression of particular histone modification enzymes. For example, heart and lymph nodes express a high variety of enzymes affecting histone acetylation and methylation, while vascular tissue does not have high expression of many modification enzymes. This may indicate that the importance of histone modification is tissue-specific. Some histone modification enzymes are more tissue-specific than the others and only a minority of them are highly expressed in multiple tissues in humans. The majority of histone modification enzymes plays an important physiological role in tissue-specific manner. Tissue differentiation signals change the expression levels of histone modification enzymes. As a response to metabolic diseases and pathological stimuli many of the enzymes are downregulated while only a few become upregulated [144]. Various histone-modifying enzymes and their marks have been implicated in the pathogenesis of cardiovascular disease [81]. Histone modification patterns are altered in AD, e.g., histone acetylation is reduced in AD brain tissues [129].

Yeast studies have shown that the impact of SAM levels on histone modifications can be site-specific because various histone methyltransferases differ in their SAM affinity. For instance, yeast Dot1 catalyzing core histone methylation is less sensitive to disrupted Met and SAM biosynthesis than histone tail methylation catalyzed by SET enzymes. The site most sensitive to SAM is H3K4me2/3. Histone methyl transferase Set1, which methylates H3K4 in yeast, acts in a multi-protein complex that comprises pyruvate kinase, serine metabolic enzymes, and SAM synthetases, and contributes to H3T11 phosphorylation and H3K4 methylation [145] (reviewed in [146]). 

Cell culture studies show that HHcy alters the expression of enzymes that modify histones. For example, in HUVECs HTL whose levels increase in HHcy [147], affects the expression of 113 genes, of which 19 encoded histones, histone-modifying enzymes, and chromatin-binding proteins (Table 3) [38]. The HTL-affected genes include histone lysine methyltransferase genes *SEDT2*, *SEDT7*, *EZH2*, *EHMT1*, *EHMT2* (up-regulated), and *SUV420H2*, *DOT1L*, *SMYD3* (down-regulated). Histone lysine acetyltransferase (*EPC1*, *EP300*) and ubiquitintransferase (*CBX2*) genes were up-regulated, while methyl-CpG binding domain protein (*MBD3*) and lysine-specific demethylase (*JMJD2B*) genes were down-regulated. Another gene, encoding regulator of histone methylation protein Jumonji (*JARID2*) is up-regulated by HTL. Bioinformatic analyses identified “Chromatin organization” as the top molecular pathway affected by HTL. The top-scored (score = 40) network was “Cardiovascular disease, cardiac infarction, skeletal system development and function” included 34 genes. HTL shows strong interactions with histone/chromatin modification and transcription genes (Figure 2).

Taken together, these findings suggest that epigenetic mechanisms involving histone modification contribute to endothelial dysfunction induced by HTL. Notably, the histone-related genes affected by HTL were not affected by Hcy or *N*-Hcy-protein, each of which changed the expression of different sets of genes [38]. 

HHcy has also been shown to significantly increase HDAC activity in human retinal endothelial cells (HRECs), human retinal pigmented epithelial cells (ARPE-19)*,* and Cbs-deficient mice retinas [65]. On the other hand, an animal study has shown that HHcy causes inhibition of HDAC3 activity [148].

### 4.3. Histone Modifications and HHcy 

Dietary factors that influence tHcy level and SAM/SAH ratio, e.g., high Met and low folate and cobalamin diets, may influence methylation rate [15]. Overall, dietary factors have been estimated to account for about 30% of serum Met variation. Other 30% of the variation was explained by clinical variables including gender and age, and the remaining unaccounted variance was likely due to genetic factors [149]. These factors affect histone modifications (Table 4). 

Acetylation and methylation 

In rodents, Met restriction (0.12% Met diet in mice) or excess (regular diet plus 1% Met in rats) has been reported to lower the SAM/SAH ratio and H3K4me3 [149]. HHcy induced by diet (high-Met plus B vitamin-deficient) in Wistar rats decreased the level of H3R8me2a in brain but not in liver and heart. The levels of other two histone modifications, namely H3R17me2a and H4R3me2a were not changed in either of the three organs [152]. Cbs-deficient mice exhibit decreased level of H4R3me2a in liver but not in brain, while other analyzed histone modifications (H3R8me2a, H3R8me2s, H3R17me2a, H4R3me2s) are not altered [28].

H3K4me3 is a histone methylation mark that encodes information e.g., about active transcription. An elegant study of the effects of Met restriction, conducted in mice and human cancer cell lines, has shown that the location of H3K4me3 peaks is largely preserved, while H3K4me3 peak width encodes changes in expression levels. It has also been shown that most of H3K4me3 peaks (81%) covers promoter regions, while a smaller subset of peaks (19%) appears on non-promoter regions such as intergenic regions and introns. Met restriction for 12 weeks in C57BL/6 adult mice alters Met metabolism, improves metabolic physiology in liver, extends life-span, and reduces global levels of H3K4me3 in vivo, while the distribution of H3K4me3 is maintained (84% promoter peaks and 16% non-promoter peaks). These findings indicate that Met availability changes metabolism via H3K4me3 width dynamics which predicts differential gene expression [153].

HHcy is associated with significantly higher neural loss in a rat model of global forebrain ischemia. Ischemia-reperfusion injury (IRI) causes massive neural disintegration in primary motor cortex region and in the cornu ammonis 1 area of the hippocampus. IRI and HHcy induce significant changes in the acetylation at Lys 9 and 12 of histones H3 and H4 in the brain cortex. The differences in histone acetylation patterns in the cortical region may have a role in pathological processes induced by IRI associated with HHcy [151].

In Cbs^+/−^ mice fed with an high-Met diet HHcy causes inhibition of HDAC3 activity and elevation of histone acetylation of H3K27ac at the promoters of IL-6 and TNF-α. This suppresses osteogenic gene expression and upregulates inflammatory cytokines. Deregulation of HDAC activity causes acetylation of NF-κB and increased NF-κB transcription factor activity in bone marrow mesenchymal stem cells of *Cbs*^+/−^ mice [148].

Clinical studies have shown an increased plasma tHcy level in patients with nonalcoholic fatty liver disease (NASH) [13,14]. In a Stelic animal model, NASH and NASH-related liver carcinogenesis are associated with dysregulation of one carbon metabolism, manifested by reduced expression of key one carbon metabolism genes, including the *AHCY* gene, which elevates SAH levels. The inhibition of *AHCY* expression is mediated by epigenetic mechanism involving gene-specific DNA hypermethylation and enrichment of the gene promoter occupancy by H3K27me3 and deacetylated histone H4K16 [75]. Diminished AHCY activity causes depletion of adenosine, which activates the DNA damage response, leading to cell cycle arrest, decreased proliferation, and DNA damage in hepatocellular carcinoma cells [37]. 

*N*-homocysteinylation

*N*-Homocysteinylation of protein lysine residues by HTL is an emerging post-translational modification, originally discovered in human non-histone proteins [32]. Recent findings show that *N*-homocysteinylation involves histone lysine residues [3,19] (Table 4) and plays a role in the development of neural tube defects (NTDs), which are known to be associated with maternal HHcy [36]. Mass spectrometry analysis of proteins from human embryonic NTD brain tissue reveals 39 histone *N*-homocysteinylation sites. Human fetal NTD brains show elevated tHcy, histone total KHcy, and specific H3K79Hcy levels. The increase in H3K79Hcy level down-regulates the expression of neural tube closure-related genes including *Cecr2*, *Smarca4*, and *Dnmt3B* in HTL-treated mouse NE4C cells. Human NTD brain tissues also show elevated tHcy, H3K79Hcy, and reduced expression of *CECR2*, *SMARCA4*, and *DNMT3B* genes. These findings show that HHcy is involved in the NTD development via up-regulation of the H3K79Hcy histone modification, leading to abnormal expression of neural tube closure-related genes [36]. As H3K79 is commonly associated with gene activation, *N*-homocysteinylation of this residue prevents gene activation. 

Protein *N*-homocysteinylation has also been shown to be involved in colorectal cancer (CRC). High-fat diet, a major risk factor for CRC, elevates plasma tHcy, which is also linked to cancer [9]. Recent study shows that high-fat diet increases *N*-Hcy-protein modification in mice and in human CRC patients [8]. Notably, the *N*-homocysteinylation involves several proteins involved in DNA damage repair. One of those proteins is ataxia telangiectasia and Rad3-related protein (ATR), a Ser/Thr protein kinase involved in DNA damage sensing. ATR interacts with HDAC2, suggesting a connection between DNA damage checkpoints and chromatin remodeling [156]. The extent of ATR *N*-homocysteinylation, correlated with Hcy levels in HCT116 and A549 adenocarcinoma cells, reduces ATR activity by impairing the ATR-ATRIP interaction, which results in DNA damage in CRC. Notably, attenuating the KHcy modification by inhibiting HTL synthesis catalyzed by MARS reduces the oncogenicity of a high-fat diet [8]. 

Recent studies indicate that protein *N*-homocysteinylation, which depends on Hcy concentration and the MARS enzyme [9,147,157], is a mechanism observed in brain aging and may have a role in pathogenesis of AD [156]. *N*-Homocysteinylated tau and MAP1 are increased and accumulated in protein aggregates and tangles in the cortex, hippocampus and cerebellum of AD and vascular dementia patients. *N*-Homocysteinylation of tau and MAPs is associated with the dissociation of tau and MAPs from β-tubulin. These results were confirmed in CA1 layer of hippocampus, cortex and cerebellum of HHcy animal models (adult rats subjected to a diet deficient in folate and vitamin B_12_, rats born to mothers deficient in vitamin B_12_ and folate during gestation and lactation, and Cd320 KO mice with selective B_12_ brain deficiency (lacking brain transcobalamin receptors)). These animal models allowed to conclude that the adverse effects of *N*-homocysteinylation accumulate with time starting in utero and increasing dramatically during aging. In cultured neuroprogenitor cells, *N*-homocysteinylation of tau and MAP1 have been shown to be dependent on MARS, as originally shown in other biological systems [147,158]. The adverse effect of *N*-homocysteinylation has been emphasized by the identification of specific Lys residues of MAPs targeted by modification with HTL, which are critical for the interactions of MAPs with β-tubulin and PSD95 [159].

### 4.4. Crosstalk between DNA Methylation and Histone Modification

There is a crosstalk between DNA methylation and histone modifications [88]. DNA methylation is integrated with histone modification status via a couple of interactions. For example, the activity of DNMTs can be stimulated by binding to certain histone lysine residues and DNA methylation may modulate histone modifications. The ADD domain of DNMT3A and DNMT3B binds to the *N*-terminal histone 3 tail unmethylated Lys 4 (H3K4), which stimulates the enzyme activity [160,161,162,163]. H3K36me3 also plays an essential role in the maintenance of a heterochromatic state by recruiting DNMT3A [86]. Another type of interaction between DNA methylation and histone modification is mediated via other molecules. For example, DNA methylation at the promoter region of a gene can modulate transcription by recruiting methylcytosine-binding proteins (MBPs), which then attract HDACs to bind to the methylated promoters. HDACs remove the acetyl group from histones, which leads to chromatin condensation and inhibits the access of regulatory proteins to the promoter [164]. 

There are several reports showing that HHcy influences gene expression via both DNA methylation and histone modification. The interaction between H3K27me3 and DNA methylation in the mouse liver, regulates *CFRT* expression. Cystic fibrosis transmembrane conductance regulator (CFTR) is a regulator of autophagy. Dysregulation of autophagy by HHcy can lead to liver injury. *Cbs*^+/−^ mice fed with a standard diet, which develop mild HHcy (22 μM tHcy) after 12 weeks, and HL-7702 cells treated with Hcy (50–500 μM) show upregulated autophagy and aggravated liver injury via down-regulation of *CFTR* expression. *CFTR* expression is modulated by the interaction of DNMT1 with histone-lysine N-methyltransferase enzyme, enhancer of zeste homolog 2 (EZH2), which regulate DNA methylation and histone H3K27 trimethylation (H3K27me3), respectively [64].

DNA hypomethylation of gene promoters induced by Hcy may lead either to down- or to up-regulation of gene expression. In HUVECs, treatments with Hcy + Ado + Ado deaminase inhibitor reduced cyclin A expression via epigenetic mechanism involving inhibited DNMT1 activity, reduced methylation of cyclin A promoter, diminished binding of methyl CpG binding protein 2 (MeCP2), and increased binding of acetylated histone H3 and H4 in the cyclin A promoter [53]. CpG hypomethylation is closely linked with histone acetylation which leads to an open chromatin and gene transcription. 

The expression of SHC-transforming protein 1 (p66shc) promotes oxidative stress and depends on promoter methylation [52]. HHcy stimulates *p66shc* transcription in HUVECs by reducing methylation of the *p66shc* promoter via inhibition of DNMT3B. Hcy treatment also increases acetylation of histone 3 on Lys 9 and 14, a signature epigenetic marks that indicate an open transcriptionally active chromatin. Thus, hypomethylation of CpG dinucleotides in the *p66shc* promoter by Hcy is associated with hyperacetylation of H3 on the *p66shc* promoter [52]. In humans, leukocyte DNA methylation at CpG 6,7 in the *p66shc* promoter is lower in individuals with elevated plasma Hcy suggesting that this mechanism operates also in vivo [52]. Reduced *p66shc* promoter methylation and elevated Hcy are also found in patients with end-stage renal disease [165].

Another example of interaction between DNA methylation and histone acetylation, emerges from studies of the mechanisms underlying schizophrenia development. Schizophrenia is associated with increased tHcy level [6]. Reelin, an extracellular matrix serine protease that plays a role in layering of neurons in the cerebral cortex and cerebellum and is important for the modulation of cell adhesion, is down-regulated in cortical interneurons of schizophrenia patients [166]. Met treatment (1 g/kg, 6.6 mmol/kg s.c. injected twice a day for 15 days), increases SAM and SAH in the brain, and decreases reelin and glutamic acid decarboxylase 67 (GAD_67_) mRNAs in both WT and heterozygous reeler male mice (B6C3Fe strain that expresses a normal and a defective reelin allele with a deletion of ~150 kb at the 3′ end of the gene (Edinburg mutation)) [167]. This effect of Met treatment is associated with an increase in the number of methylated cytosines in the CpG island of the reelin promoter region. 

The treatment with valproic acid (300 mg/kg, 2 mmol/kg for 15 days, twice a day), an inhibitor of HDACs, increases acetylation of histone H3 in mouse brain and reverts Met-induced down-regulation of reelin and GAD_67_ in both WT and heterozygous reeler mice. 

The molecular changes in response to Met treatment in WT and reeler mice are associated with behavioral defects. Prepulse inhibition of startle (PPI) enables the organism to filter out the unnecessary information. PPI deficits are observed in patients with schizophrenia and AD and is thought to be an indicator of sensory gating deficit. In WT and heterozygous reeler mice Met treatment accelerates the decline of PPI occurring with increase of the delay in the prepulse/startle intervals indicating that the hypermethylation of promoters may be a plausible mechanism of schizophrenia development [168]. 

The down-regulation of Reelin and GAD_67_ expression is likely to be linked with hypermethylation of their promoters caused by increase of DNMT1 in GABAergic neurons. Treatment of mice with valproate elevates histone 3 acetylation, prevents Met-induced reelin promoter hypermethylation, reelin mRNA down-regulation, and PPI and social interaction deficits, normalizing behavioral effects induced by Met [169]. The benefits of valproate treatment may arise from the inhibition of histone deacetylase and activation of brain DNA demethylation [170].

The down-regulation of Reelin and GAD67 expression due to their promoter hypermethylation upon Met treatment leads to recruitment of transcription repressor proteins [methyl-CpG binding protein (MeCP2) and HDACs], which results in transcriptionally inactive chromatin. These changes are reversible and modulated by HDAC inhibitors which activate DNA-cytosine-5-demethylation. Levels of RELN and GAD_67_ promoter hypermethylation induced by a 7-day Met treatment declines by 50% after 6 days of Met withdrawal. Valproate accelerates RELN and GAD_67_ promoter demethylation and increases the binding of acetylated histone 3 to RELN and GAD_67_ promoters. These findings suggest that histone modifications modulate DNA demethylation and that HDAC inhibitors affect DNA demethylation [171].

## 5. HHcy and Noncoding RNA Regulation

Genome-wide analyses and RNA-sequencing technologies enabled the discovery of noncoding RNAs, such as microRNAs (miRNAs), linear long noncoding RNAs (lncRNAs), and circular long noncoding RNAs (circRNAs). Preponderance of evidence indicates that ncRNAs are important in the etiology of diseases including those associated with HHcy. 

### 5.1. MiRNAs

MiRNAs are 21–22-nucleotide small noncoding RNAs that negatively regulate their target mRNAs by binding to regulatory elements located in the 3′ -untranslated region (3′ -UTR) of the mRNA [172]. MiRNAs control the expression of genes important for cellular processes such as proliferation, apoptosis, differentiation, and the development of diseases [173,174]. The dysregulation of miRNA can be triggered by multiple mechanisms, including aberrant DNA methylation of the miRNA promoter (15) and is implicated in disease development. By controlling the function of macrophages, ECs, and VSMCs miRNAs play an important role in the progression of atherosclerosis [175,176]. 

Normal physiological function of blood vessels is maintained by VSMCs, which comprise most of the blood vessel wall. The main function of VSMCs relies on changing the volume of blood vessels through vasoconstriction and vasodilation. Their excessive proliferation contributes to pathological states including atherosclerosis. Moreover, VSMCs dynamically change their structure and function during plaque formation. The switch between differentiated (contractile) and dedifferentiated (proliferative, migratory and synthetic phenotype) phenotypes of VSMCs is required during vascular remodeling in response to blood vessel injury, stress, and inflammation and occurs as a response to pathological stimuli and biochemical regulators, among which miRNAs play an important role [177]. One of the most abundant miRNA in the vascular wall is miR-145, which forms a small cluster with miR-143 [178]. The function and regulation of miR-145 in vascular disease have been the focus of many recent studies [179]. miR-145 serves as a VSMC phenotypic marker and modulator in controlling vascular neointimal lesion formation. As certain miRNA may exert its influence on more than one target gene, it is believed that miR-145 is involved in different diseases via different pathways [180].

Apart of miR-145, other miRNAs, including miR-143, miR-221/222, miR-24, miR-146a, and miR-21, have been found to modulate VSMC differentiation, phenotypic switch and neointimal formation following vascular injury [181,182]. Among these, miR-143 is a markedly enriched in VSMCs and has been investigated extensively. The expression of miR-143 is critical to the modulation of VSMCs phenotype, by promoting their differentiation and repressing proliferation [177].

In addition to promoting atherosclerosis via alteration of DNA methylation, HHcy can also be proatherogenic via its effects on miRNAs in VSMCs. For example, HHcy induced in cultured VSMCs and also in vivo in the *ApoE*^−/−^ mice, reduces miR-125b expression, which upregulates DNMT3B. This leads to hypermethylation and reduced expression of the *p53* gene, thereby promoting VSMC proliferation [60]. Another study shows that HHcy upregulates DNMT3A expression (mRNA and protein), increases methylation and downregulates the *miR-143* gene expression in VSMCs [61].

HHcy by changing chromatin structure provokes cardiac remodeling. In HHcy mice cardiac-specific deletion of *N*-methyl-d-aspartate receptor-1 (NMDAR1) ameliorates the Hcy-induced alteration in calcium transient, restores contractility of cardiomyocytes [183], and mitigates mitophagy induced by matrix metalloproteinase-9 (MMP9) [184]. Further studies have shown that epigenetic mechanisms are involved. HHcy induced in cultured HL-1 cardiomyocytes increases the expression of NMDAR1, DNMT1, and matrix MMP9, and H3K9 acetylation, while decreases levels of HDAC1, miR-133a, and miR-499. Similar observations were made in heart tissue of *Cbs*^+/−^ mice [63]. HHcy enhances the expression of dicer, and 11 miRNAs, and downregulates miR-188 expression in cultured HL-1 cardiomyocytes. Additionally, HHcy cardiomyocytes exhibit elevated levels of MMP-2,-9 and TIMP-1,-3, and reduced expression of TIMP-4, as well as increased NOX-4 expression, a marker of oxidative stress [185].

Treatment of HL-1 cardiomyocytes with Hcy upregulates Cse, downregulate Cbs and miR-133a in a dose-dependent manner, and induces hypertrophy in murine atrial. Treatment with Na_2_S/GYY4137, a H_2_S donor, downregulates the specificity protein-1 (SP1), an inducer for Cse, and upregulates miR-133a that targets SP1 and inhibits cardiomyocytes hypertrophy. In HHcy Cbs-deficient mice, cardiac Cse is upregulated, probably due to up-regulation of SP1 [186].

HHcy induced by a high-Met diet in *ApoE*^−/−^ mice upregulates expression of miR-148a/152, and decreases DNMT1 mRNA and protein levels in the aorta. Similar changes ae observed in cultured foam cells stimulated with Hcy. On the contrary, DNMT1 overexpression enhances DNA methylation and reduces miR-148a/152 expression. The cross-talk between miR-148a/152 and DNMT1 in foam cells, may play a critical role in HHcy-related atherosclerosis (Table 2) [56].

MiRNA profiling in mouse retinas identified 127 miRNAs in *Cbs*^+/–^ and 39 miRNAs in Cbs^–/–^ mice whose expression was significantly changed compared to *Cbs*^+/+^ animals. Pathway analysis shows the involvement of these miRNA in HDAC and DNMT activation, endoplasmic reticulum and oxidative stresses, inflammation, hypoxia, and angiogenesis. HHcy-induced epigenetic modifications may be involved in retinopathies associated with HHcy, such as age related macular degeneration and diabetic retinopathy [65].

HHcy is associated with decline in muscle function and it has been shown that HHcy plays a causal role in enhanced fatigability through mitochondrial dysfunction which involves epigenetic changes. HHcy *Cbs*^+/−^ mice exhibit more fatigability, generate less contraction force and have reduced large muscle fiber number. Increased fatigability is not caused by changes in key enzymes involved in metabolism, but rather by reduced ATP levels in the skeletal muscle. Cbs^+/−^ mice have significantly reduced dystrophin and mitochondrial transcription factor A (mtTFA) levels. There are specific microRNAs (miR-31, miR-494, and miR-499) that are expressed in skeletal muscles and have been shown to influence either dystrophin levels or modulate mitochondrial function. Cbs^+/−^ mice show elevated miR-494 and miR-31 levels. Mutant mice have significantly decreased level of the transcriptional mtTFA regulator NRF-1. Exercise reverses the molecular changes induced by HHcy, with the exception of dystrophin levels. HHcy in Cbs^+/−^ mice affects mtTFA expression at both the transcriptional (via reduced NRF-1 level) and post-transcriptional levels (via increased miR-494 level). Myoblast cells C2C12 treated with Hcy have increased miR-494 levels and decreased mtTFA protein quantity. Additionally, in C2C12 cells HHcy elevates de novo methylating enzymes (DNMT3A and DNMT3B) and global DNA methylation levels [69].

HHcy affects blood-brain barrier (BBB) integrity via upregulating matrix metalloproteinases (MMPs), that play a role in matrix remodeling and lead to blood–endothelial barrier leakage [187]. Hcy may cause BBB dysfunction by affecting DNA and histone methylation, and miRNA levels. It has been reported that BBB integrity is mediated by miR29b through regulating DNMT3B, which modulates the levels of MMP9, that digest the membrane and junction proteins leading to leaky vasculature. Treatments with Hcy (50–200 μM) elevate miR29b level as well as MMP9 expression and activity in mouse brain endothelial cells. HHcy in *Cbs*^+/−^ mice, accompanied by high brain vessel permeability, is also associated with elevation of miR29b as compared with wild-type animals [68]. Male *Cbs*^+/−^ deficient mice fed a high Met diet have decreased MTHFR expression and increased AHCY expression, elevation of DNMT1, DNMT3A, up-regulation of matrix metalloproteinases (MMP-2, MMP-9), a decrease in the tissue inhibitors of metalloproteinases (TIMP-1 and TIMP-2), lower expression of tight junction proteins (ZO-1, ZO-2, occludin), increased permeability of the blood-brain barrier, neurodegeneration, and synaptotoxicity. In addition treatment with folic acid of *Cbs*^+/−^ mice fed with high-Met diet results in a decrease in brain tHcy and rescues neurotoxic alterations caused by Met supplementation [74].

MiRNAs regulate also BBB and play a role in maintaining its integrity. In mouse brain endothelial cell line bEnd.3 HHcy induces a several-fold increase of about 20 miRNAs, with the highest elevation in miR29b family members (miR29a, miR29b1, miR29b2, and miR29c), miR298, miR33, miR431, and miR381. MiR29b regulating DNMT3B and MMP9 levels, may lead to BBB disruption. Results of cell culture studies were confirmed in Cbs^+/−^ mouse model, characterized by high brain vessel permeability and elevated miR29b levels [68].

### 5.2. LncRNAs

HHcy may also influence the expression of H19 and insulin growth factor 2 (IGF2). The *H19* gene codes a 2.3 kb RNA product, which is capped, spliced, and polyadenylated, but probably is not translated [188]. H19 is a multifunctional lncRNA located predominantly in the cytoplasm but found also in the nucleus. It regulates gene expression via various mechanisms [189]. One of the suggested functions of H19 RNA is to regulate the expression of IGF2 [190]. Methylation of the *H19* promoter decreases H19 expression and increases the expression of IGF2, a neighboring gene on chromosome 11 [191]. These two imprinted genes (*H19* and *Igf2*) are positioned in close proximity on mouse chromosome 7 and their expression is regulated by methylation of a differentially methylated domain (DMD) located 5′ to *H19*. In a mouse model of HHcy (Cbs^+/−^ and C57BL/6 Cbs^+/+^ mice fed a HHcy or control diet, from weaning until 9-12 months of age, HHcy diet contained double the amount of Met, minimal folic acid, and lower choline, pyridoxine, cobalamin, and riboflavin) higher plasma tHcy was accompanied by higher levels of SAH and lower SAM/SAH ratios in the liver and brain. Interestingly, HHcy influenced *H19* DMD methylation in a tissue-specific manner. HHcy mice had decreased *H19* DMD methylation in the liver, while the methylation increased in the brain and aorta. Increased *H19* DMD methylation was accompanied by increased expression of H19 transcripts. Additionally, levels of H19 transcripts in aorta correlated positively with plasma tHcy. Summing up, HHcy causes tissue-specific changes in *H19* DMD methylation and increased vascular expression of H19 in adult mice [192]. 

The toxicity of HHcy is often studied in a *Cbs*^+/−^ mouse model fed with an HHcy diet. Such animals have elevated plasma tHcy, liver SAH, and reduced SAM/SAH ratios. The changes in Hcy metabolites are accompanied by lower liver maternal *H19* DMD allele methylation, decreased liver Igf2 mRNA levels, and loss of *Igf2* maternal imprinting. There is a negative correlation between liver SAH level and liver maternal *H19* DMD methylation in all mice. On the other hand, no changes in SAM and SAH between HHcy and control mice are detected in the brain but HHcy mice exhibit higher maternal *H19* DMD methylation and lower H19 mRNA levels in the brain [97].

H19 may regulate gene expression via its interaction with AHCY. It has been shown that H19 binds AHCY and inhibits its ability to hydrolyze SAH to Hcy and adenosine in human and mouse cells. As a result, SAH inhibits SAM-dependent methyltransferases including DNMTs, which alter level of methylation of some genomic loci. In this way H19, by modulating the activity of AHCY, may regulate methylation at many genomic sites and affect epigenetic landscape in a genome-wide manner [193].

### 5.3. CircRNAs

CircRNAs are generated by back splicing of specific regions of pre-messenger RNAs. circRNAs have miRNA binding sites, regulate gene expression through the sponging of miRNA, and are abnormally expressed in human disease [194], including myocardial infarction, atherosclerosis, cardiomyopathy, and cardiac fibrosis [195]. 

Recent findings suggest that HHcy pathophysiology involves circRNA (178). CBS deficiency in humans leads to severe HHcy, which causes retinovascular thromboembolism, eye-lens dislocation, vascular cognitive impairment, and dementia. HHcy results in vascular dysfunction, inflammation, and redox imbalance leading to neurovascular pathologies. Analysis of circRNAs in the eyes of Cbs^+/−^ mice identified a pool of 12,532 circRNAs, 74 of which exhibited differential expression (70% up- and 30% down-regulated). Altered circRNAs profile can affect disease phenotype by affecting miRNAs levels, which in turn affect gene expression leading to redox imbalance, internal inflammation, and mitochondrial dysfunction [196]. HHcy induced in human retinal pigment epithelial (RPE) cells (ARPE-19), changes expression of circRNAs. Out of 12,233 identified circRNAs, 54 were differentially expressed (17 were down-regulated, and 37 were up-regulated) [197].

## 6. Summary/Prospects

Elucidation of mechanisms of HHcy toxicity is crucial for prevention and treatment of major human diseases, including atherosclerosis, cancer, dementia and brain disease, liver injury, osteoporosis, and pregnancy complications. Recent findings have revealed that the pathogenesis of many diseases, including those induced by HHcy, involves dysregulation of epigenetic mechanisms that control cell growth and differentiation. Details of the molecular and cellular mechanisms involving interactions of HHcy with DNA methylation, histone modifications, miRNA, and lncRNA are beginning to emerge. The failure of Hcy-lowering trials to reduce heart attacks, suggests that some pathological changes caused by HHcy are irreversible and highlights the need for additional research into the mechanisms by which HHcy causes disease. It is likely that epigenetic dysregulation induced by HHcy cannot be reversed by dietary interventions targeting Hcy. In fact, folic acid supplementation, which lowers plasma Hcy levels and elevates SAM/SAH ratio, does not affect global DNA methylation in HHcy subjects with normal renal function [198,199]. Although, in patients with uremia, folate treatment corrects global DNA hypomethylation [115], this is not always associated with better outcomes [200]. Folic acid supplementation also does not lower HTL [25] and anti-*N*-Hcy-protein autoantibodies [201] in CAD patients. Accumulating evidence suggests that targeting HTL, which is involved in the pathology of HHcy [19,25], might be an effective strategy for disease prevention. One possible therapeutic approach is to pharmacologically inhibit MARS using Hcy analogues, which reduce HTL synthesis at the MARS active site, which in turn attenuates protein *N*-homocysteinylation [8]. Another approach would be to up-regulate HTL-hydrolyzing enzymes, such as PON1, BLMH, or BPHL, which also would reduce HTL levels and *N*-Hcy-protein accumulation [202,203]. Although epigenetic dysregulation in HHcy has been proposed to be secondary to the accumulation of SAH, recent evidence indicates that in general SAH *may not be* associated with DNA hypomethylation. Post-translational modification of histone lysine residues by HTL has been established as a new epigenetic mechanism responsible for neural tube defects. Analogous modifications of other proteins by HTL are involved in other diseases such as atherosclerosis, colorectal cancer, and Alzheimer’s disease. Additional studies are required to determine how general these mechanisms are and whether they can explain the pathology of other HHcy-related diseases.

## Figures and Tables

**Figure 1 ijms-20-03140-f001:**
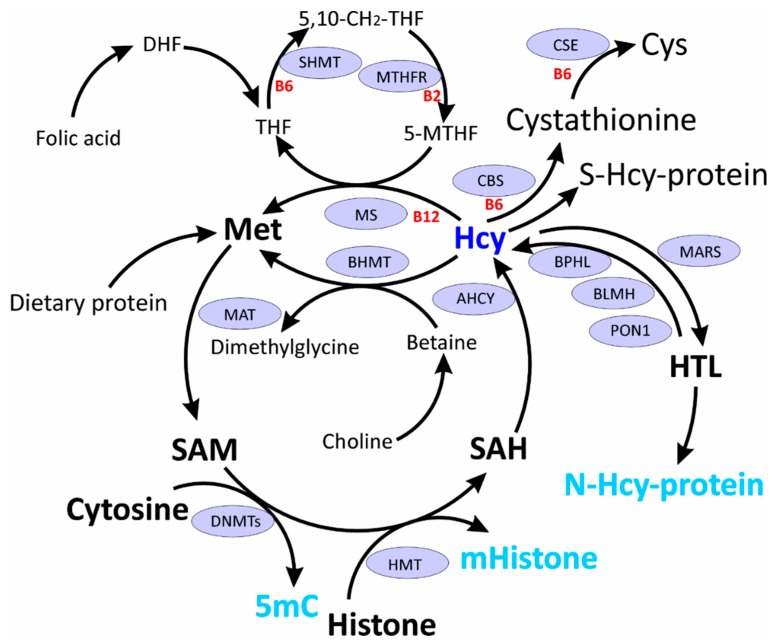
Homocysteine (Hcy) metabolism. See text for description. Metabolites and enzymes related to epigenetics are highlighted in blue. AHCY, S-adenosylhomocysteine hydrolase; BHMT, betaine:Hcy methyltransferase; BLMH, bleomycin hydrolase; BPHL, biphenyl hydrolase like; CBS, cystathionine β-synthase; CSE, cystathionine γ-lyase; DNMTs, DNA methyltransferases; HMT, histone methyltransferase; HTL, homocysteine thiolactone; MARS, Met-tRNA synthetase; mHistone, methylated histone; MTHFR, 5,10-methylenetetrahydrofolate reductase; MS, Met synthase; PON1, paraoxonase 1; SAH, S-adenosylhomocysteine; SAM, S-adenosylmethionine; SHMT, serine hydroxymethyltransferase.

**Figure 2 ijms-20-03140-f002:**
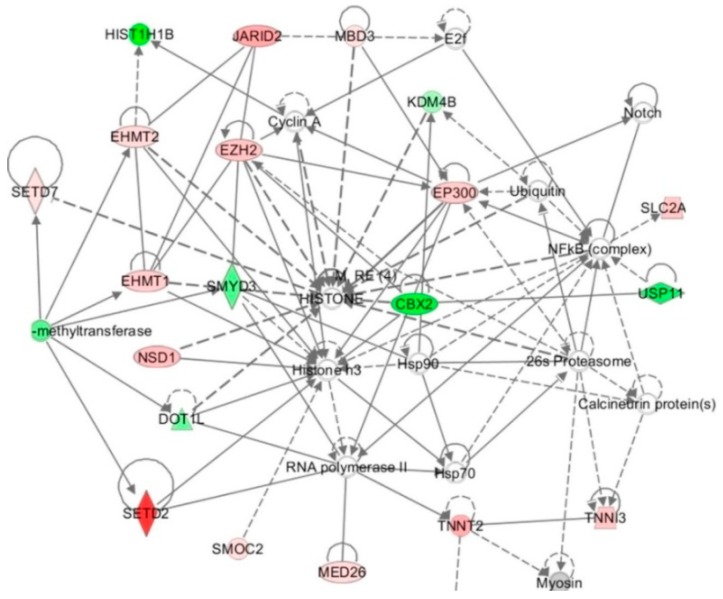
The top network of HTL-responsive genes: Cardiovascular disease, cardiac infarction, skeletal and muscular system development and function. The major nodes in the network contain histones and other chromatin-related proteins (reproduced with permission from reference [38]).

**Table 1 ijms-20-03140-t001:** Effects of hyperhomocysteinemia (HHcy) on DNMTs, SAM, SAH, SAM/SAH, and Promoter DNA Methylation in vitro.

Cell Line/Model Organism	Treatment	Effect on			Reference
		SAM, SAH, SAM/SAH	DNMT	*Promoter DNA* Methylation/Gene Expression mRNA/Protein	
**HUVEC**	Hcy 1 mM, 5 days	NA	NA	↓genomic DNA methylation ↑SORBS1/↓/↓	[47]
**HUVEC**	Hcy 10, 30 μM, 72 h Hcy 100, 300 μM, 72 h	NA NA	NA NA	↓*DDAH2*/↑/NA ↑*DDAH2*/↓/NA	[48]
**HUVEC**	Hcy 1 mM, 48 h	NA	↑DNMT1 protein	↑*DDAH2*/↓/NA	[49]
**HUVEC**	Hcy 100, 200, 500 μM, 72 h	NA	↓DNMT1 mRNA, protein	↓*LOX-1*/↑/↑	[50]
**HUVEC**	Hcy 50, 100, 200 μM, 24 h	NA	↓DNMT1 mRNA, protein ↓DNMT activity	↓*PDGF-A, -C, -D*/↑/NA	[51]
**HUVEC**	Hcy 200 μM, 8 h	NA	↑DNMT3B ↓DNMT activity	↓*p66shc*/↑/↑	[52]
**HUVEC**	D,L-Hcy 50 μM, Ado 40 μM, 10 μM Ado-deaminase inhibitor, 48 h	NA	↓DNMT1 activity =DNMT3 activity	↓*Cyclin A*/↓/NA	[53]
**HUVEC**	Hcy 25, 50, 100, 200 μM, 24, 48, 72 h	NA	NA	↓*sEH*/↑/↑ mRNA	[54]
**HUVEC**	Hcy 25, 50, 100, 200 μM, 72 h	NA	NA	↓hTERT/↓/↓	[55]
**Human foam cells**	Hcy 50, 100, 200, 500 μM, 48 h	NA	↓DNMT1 mRNA, protein	NA/↓*miR-148a/152*/NA	[56]
**Human foam cells**	Hcy 100 μM, 72 h	NA	NA	↓*FABP4*/↑/↑	[57]
**Human monocytes**	Hcy 50, 100, 200, 500 μM, 48 h	↓SAM, ↑SAH ↓SAM/SAH	↑DNMT activity	↑*PPAR**γ*/↓/↓	[58]
**T/G HA** **-** **VSM**	Hcy 50, 100, 200 and 500 μM, 72 h	NA	↑DNMT1 mRNA, protein	↑*PTEN*/↓/↓	[59]
**VSMC**	D,L-Hcy 50, 100, 200, 500 μM, 72 h	NA	↑DNMT3B protein	↑*p53*/↓/↓	[60]
**VSMC**	Hcy 50, 100, 200, 500 μM, 72 h	NA	↑DNMT3A mRNA, protein	↑/↓miR-143/NA	[61]
**VSMCs**	Hcy 50, 100, 200 and 500 μM, 72 h	NA	↑DNMT1 mRNA, protein	↑MFN2/↓/↓	[62]
**HL-1 cardiomyocytes**	Hcy 5, 100 μM, 72 h	NA	↑DNMT1 mRNA, protein	NA	[63]
**Human hepatocytes (HL-7702)**	Transfection with Ad-CFTR L-Hcy 100 μM, 24 h	NA	NA	↑*CRFT*/↓/↓	[64]
**Human primary retinal endothelial cells (HRECs)**	Hcy 20, 50, 100 μM	NA	↑DNMT activity	NA	[65]
**Human retinal pigments epithelial cells (ARPE-19)**	Hcy 20, 50, 100 μM	NA	↑DNMT activity	NA	[65]
**Mouse *Cbs*^−/−^, *Cbs*^+/−^ retinas**	None	NA	↑DNMT activity	NA	[65]
**Mouse endothelial progenitor cells** **differentiated** **from primary bone marrow mononuclear cells isolated from femur and tibia of female C57BL/6J mice**	High-Met, low-folate, low-vitamin B_6_ & B_12_ diet 8 weeks	NA	↑DNMT1 mRNA, protein =DNMT3A mRNA, protein ↑DNMT activity	↑global DNA methylation ↑*Cbs*/↓/↓	[66]
**Mouse brain endothelial cells** **(bEnd.3)**	Hcy 100 μM, 24 h	NA	↑ DNMT1 mRNA, protein ↑DNMT3A mRNA, protein ↓ DNMT3B mRNA, protein	NA	[67]
**Mouse brain endothelial cell line (bEnd.3)**	Hcy 50, 100, 200 μM, 24 h	NA	↑DNMT3A protein ↓DNMT3B protein	NA	[68]
**Mouse myoblast C2C12 cells**	Hcy 500 μM, 3 days	NA	↑DNMT3A protein ↑DNMT3B protein	NA	[69]
**Mouse neuro 2A cells, neuroblastoma cell line stably expressing human APP carrying the** **K670N, M671L Swedish mutation (N2A-APPswe)**	DL-Hcy 50 μM, adenosine 40 μM, erythro-9-(2-hydroxy-3-nonyl)-adenine hydrochloride 10 μM, 24 h	↓SAM ↑SAH	↓DNMT1 protein ↓DNMT3A protein ↓DNMT3B protein	↓*5LO*/↑/↑	[70]

**Table 2 ijms-20-03140-t002:** Effects of HHcy on SAM, SAH, DNMTs, and Promoter DNA Methylation in vivo.

Model Animal, Tissue	Treatment	Plasma/Tissue tHcy, μM	Effect on			Reference
			SAM SAH SAM/SAH	DNMT	*Promoter DNA* Methylation/Gene Expression mRNA/Protein	
**Male *ApoE*^−/−^ mice,** **Aorta**	High-Met diet 20 weeks	Plasma tHcy 2.67 ± 0.79 (control diet) vs. 13.79 ± 0.54 (*ApoE^-^*^/-^) and 6.40 ± 0.28 (*ApoE^+^*^/+^) (high-Met diet)	↑SAM ↑SAH ↑SAM/SAH	↓DNMT1 mRNA, protein =DNMT3A mRNA, protein =DNMT3B mRNA, protein	↓*FABP4*/↑/↑	[57]
**Male *ApoE*^−/−^ mice**	High-Met diet 15 weeks	NA	NA	↑DNMT3B protein	NA/↓miR-125b/NA	[60]
**Male *Tg-127T Cbs*^−/−^ mice** **3-month-old** **Brain cortices**	None	Plasma tHcy 296 vs. 5.5 (controls) [71]	NA	↓DNMT1 protein ↓DNMT3A protein ↓DNMT3B protein	↓*5LO* /↑/↑	[70]
**Male Cbs-deficient mice,** **8–12-week-old,** **Heart**	High-Met diet	NA	NA	↑DNMT1 protein ↑DNMT activity	↑Genomic DNA methylation	[63]
**Male C57BL/6J mice,** **8-week-old,** **Aortic intima**	High-Met diet 4 or 8 weeks	Plasma tHcy 27.6 ± 4.5 or 61.5 ± 31.4 vs. 5.2 ± 1.3 in the control group after 4 or 8 weeks, respectively	NA	NA	↓*PDGF-A, -C* and *-D*/↑/↑	[51]
**Male C57BL/6J mice,** **6–8-week-old** **Blood**	High-Met diet 8 weeks	Plasma tHcy 61.5 ± 31.4 vs. 5.2 ± 1.3 in the control group after 4 or 8 weeks, respectively [51]	NA	NA	↓*mTERT*/↓/NA	[72]
**Male C57BL/6J mice,** **6–8-week-old,** **Aorta**	High-Met diet 4 or 8 weeks	Plasma tHcy 61.5 ± 31.4 vs. 5.2 ± 1.3 in the control group after 4 or 8 weeks, respectively [51]	NA	NA	↓*mTERT*/↓/↓	[55]
**Male Sprague-Dawley rats,** **4-week-old** **Blood**	High-Met diet 8 weeks	Serum tHcy level 66.8±11.7	NA	NA	↓*rTERT*/↓/NA	[73]
**Male Cbs-deficient mice** **8–12-week-old** **Brain**	High-Met diet	NA	NA	↑DNMT1 mRNA, protein ↑DNMT3A mRNA, protein ↑DNMT activity	↑Genomic DNA methylation	[74]
**Male and female 3xTg-AD mice,** **Brain cortices**	Folate, vitamin B_6_, B_12_-deficient diet 7 months	NA	↓SAM ↑SAH	↓DNMT1 protein ↓DNMT3A protein ↓DNMT3B protein	↓*5LO*/↑/↑	[70]
***Cbs*^+/−^ mice,** **8–10-week-old,** **Liver**	None	3.46 times higher than in *Cbs*^+/+^ mice	NA	↑DNMT1 mRNA, protein =DNMT3A mRNA, protein =DNMT3B mRNA, protein	↑*CRFT*/↓/↓	[64]
**Male C57BL/6J mice,** **Stelic animal model,** **Liver**	Streptozotocin injection at 2^nd^ day High-fat diet 6, 12, 20 weeks	=liver Hcy ↑plasma Hcy	↑liver SAM ↑liver SAH ↓liver SAM/SAH ↑plasma SAH	↑DNMT1 mRNA, protein ↑DNMT3A mRNA, protein ↑DNMT3B mRNA, protein	↑*Ahcy*/↓/?	[75]
**Male C57BL/6J mice,** **CFD model,** **Liver**	Low-Met, w/o choline and folic acid (CFD) diet 12 weeks	↑liver Hcy ↑plasma Hcy	↓liver SAM =liver SAH	NA	NA *Ahcy*/↓/?	[76]
**Male Fisher 344 rats,** **4-week-old** **Liver,** **preneoplastic liver, and liver tumor**	Low-Met, w/o choline and folic acid diet, 36, 54 weeks	NA	NA	NA	↓LINE-1	[77]
**Male F344 rats** **4-week-old** **Liver**	Low-Met, w/o choline and folic acid diet, 9, 18, 24, and 36 weeks, followed by 18 weeks of feeding a methyl-adequate diet with sufficient content of Met, choline, and folic acid	NA	↓ SAM ↓ SAM/SAH	NA	↓global DNA methylation (reversible after 9 weeks and irreversible after 18-36 weeks of the methyl-deficient diet)	[78]
**Male F344 rats** **4-week-old** **Brain**	Low-Met, w/o choline and folic acid diet, 18, 36 weeks	↑Hcy After 36 weeks ~0.15 vs. 0.1 nmol/mg protein	=SAM =SAH =SAM/SAH	↓DNMT1 protein ↑DNMT3A protein ↑DNMT3B protein	↑DNA methylation within unmethylated GC-rich DNA domains =methylation within methylated GC-rich DNA regions	[79]
**Male Sprague-Dawley rats,** **8 weeks old**	High-Met diet Low-Met diet 4 weeks	↑plasma tHcy	↓SAM ↑SAH ↓SAM/SAH	↑DNMT3A mRNA, protein ↑DNMT3B mRNA, protein	↓genome methylation in B1 repetitive elements	[80]
**Male *Cbs*^+/−^ and *Cbs*^+/+^ mice** **8–10-week-old,** **Brain**	5′-aza (0.5 mg/kg body weight) intraperitoneally for 6 weeks, +/− Met, +/− folic acid (FA)	↑Hcy	NA	↓DNMT3B protein	↑5-mC in brain DNA from CBS^+/−^, + Met mice; tended to decrease in FA-supplemented mice	[68]
**Male *ApoE*^−/−^ C57BL6J mice,** **6-week-old, Aorta**	High-Met, low folate/B_12_ diet 20 weeks	↑Serum Hcy	NA	↓DNMT1 mRNA, protein	NA	[56]
**Male C57BL/6J *sEH*^+/+^ and *sEH*^−/−^ mice,** **8-week-old,** **Aortic intima**	High-Met diet 4, 8 weeks	↑Plasma tHcy	NA	NA	↓*sHE*/↑/NA	[54]

**Table 3 ijms-20-03140-t003:** Hcy-thiolactone (HTL) treatment affects the expression of histone and histone-related genes in HUVEC. mRNA levels were quantified using microarrays (*p* < 0.05) [38].

Gene Name	Access. No.	Protein Name	Fold Change at HTL 10 μM 1000 μM	
***SETD2***	29072	Histone-lysine N-methyltransferase SETD2	17.9	6.1
***SETD7***	80854	Lysine methyltransferase 7	1.1	3.9
***EZH2***	2146	Histone-lysine N-methyltransferase EZH2	2.3	5.4
***EHMT1***	79813	Euchromatic histone-lysine N-methyltransferase 1	2.0	3.7
***H2AFY***	9555	H2A histone family, member Y	1.7	2.5
***EHMT2***	10919	Euchromatic histone-lysine N-methyltransferase 2	1.7	2.5
***EPC1***	80314	Enhancer of polycomb homolog 1 (Histone acetylase)	1.8	2.0
***EP300***	2033	E1A-binding protein (Histone acetylase)	2.1	2.8
***HIST1H2BK***	85236	Histone cluster 1, H2bk	2.3	2.5
***JARID2***	3720	Jumonji, AT-rich interactive domin 2	3.6	2.7
***MBD3***	53615	Methyl-CpG binding domain protein 3	1.5	2.5
***SUV420H2***	84787	Histone-lysine N-methyltransferase	−1.9	−3.3
***DOT1L***	84444	Histone-Lys N-methyltransf, H3K79 specific	−2.5	−2.3
***SMYD3***	3009	Histone-lysine N-methyltransferase	−2.8	−11.3
***JMJD2B***	23030	Lysine-specific demethylase 4B	−1.9	−3.0
***BRD8***	10902	Bromodomain-containing protein 8	−2.8	−4.8
***CBX2***	12416	Chromobox prot homol 2 (H2AK119 ubiq.)	−6.0	−7.1
***HIST1H2AJ***	8330	Histone cluster 1, H2ak	−2.7	−4.1
***HIST1H1B***	3009	Histone cluster 1, H1b	−7.7	−9.4

**Table 4 ijms-20-03140-t004:** Effects of HHcy or Met restriction on histone acetylation, methylation, and *N*-homocysteinylation.

Organism/Cell Line	Treatment	Effect	References
**Acetylation**			
**HUVEC**	Hcy or HTL 1, 10, 100, 1000 μM	↓H3K9ac	[19,150]
**HUVEC**	Hcy 50 μM 24 h	↑H3K9ac ↑H3K14ac	[52]
**HL-1 cardiomyocytes**	Hcy 100 μM, 72 h	↑H3K9ac	[63]
**Human colorectal cancer cell line HCT116**	Met restriction 24 h	↓H3K9ac	[149]
**Male C57BL/6J mice** **Stelic animal model** **Liver**	Streptozotocin injection at 2^nd^ day High-fat diet from 4^th^ week of age 6, 12, 20 weeks	↓H4K16ac	[75]
**Female *Cbs*^+/−^ mice (129P2-Cbs^tm1Unc^/J)** **16-weeks old** **Femur bone**	Met-rich, low folate vitamin B_6_, B_12_ diet 8 weeks	↑H3K27ac	[148]
**Male Wistar rats** **4–6-months old** **Brain cortex**	Hcy 1.2 μmol/g of body weight injected subcutaneously once a day 21 days	↑H3K9ac ↑H4K12ac	[151]
**Methylation**			
**Male C57BL/6J mice Stelic model** **Liver**	Streptozotocin injection at 2^nd^ day High-fat diet from 4^th^ week of age 6, 12, 20 weeks	↑H3K27me3 in NASH-derived hepatocellular carcinoma	[75]
**Female Wistar rats** **Brain** **Heart** **Liver**	Met-rich, deficient in B vitamins (folic acid, B_6_ and B_12_) 8 weeks	↓H3R8me2a (brain) = H3R8me2a (heart, liver) = H3R17me2a (brain, heart, liver) = H4R3me2a (brain, heart, liver)	[152]
**Male Fisher 344 rats,** **4-weeks old,** **Liver,** **Preneoplastic liver, Liver tumor**	Low-Met and lacking in choline and folic acid diet 36, 54 weeks	↓H4K20me3 ↑H3K9me3 (preneoplatic nodules and liver tumors)	[77]
***Cbs*^+/−^ mice** **8–10-weeks old,** **Liver**	None	↑H3K27me3 = H3K27me1 = H3K27me2	[64]
***Tg-I278T Cbs*^−/−^ mice,** **Brain, Liver,** **Heart, Kidney**	None	↓H4R3me2a (liver) = H4R3me2a (brain)	[28]
**Human colon cancer cells HCT116**	Met restriction 24 h	↓H3K4me3 ↓H3K9me3 ↓H3K27me3	[149]
**Human colon cancer cells SW620, SW480, HCT8, HT29, NCI-H5087**	Met restriction 24 h	↓H3K4me3	[149]
**C57BL/6J mice** **8-week-old, Liver**	Low-Met diet 12 weeks	↓H3K4me3	[149]
**Male C57BL/6J mice 7-week-old**	Low-Met diet 12 weeks	↓H3K4me3	[153]
**Human colon cancer cells HCT116**	Met restriction 24 h	↓H3K4me3	[153]
**Human hepatocytes (HL-7702)**	Transfection with Ad-CFTR L-Hcy 100 μM, 24 h	↑H3K27me3	[64]
***N*-homocysteinylation**			
**HUVEC**	HTL or Hcy 1, 10, 100, 1000 μM; 24 h	↑*N*-Hcy-H1, -H2A, -H2B, -H3, -H4	[19,150]
**Human embryos**	NA	39 *N*-Hcy-sites: 8 in H2a, 16 in H2b, 8 in H3, 7 in H4	[36]
**Chicken embryos**	0.5 μL of 0.5 mM HTL injected into the neural groove	↑*N*-Hcy-histone	[36]
**Mouse neural stem cells (NE4C)**	DL-Hcy or L-HTL 0.1, 0.5, 1 mM 8 h	↑*N*-Hcy-histone	[36]
**HEK293 cells**	DL-Hcy or L-HTL 0.1, 0.5, 1 mM 8 h	↑*N*-Hcy-histone	[36]
**HEK293 cells**	MARS knockdown	↓ *N*-Hcy-histone	[36]
**HEK293T, HCT116,** **HeLa cells**	HTL 0.01, 0.1, 1 mM 24h	↑ H3K23Hcy ↑ H3K79Hcy	[154] [155]

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
