# Peer review of "Dysregulation of Epigenetic Mechanisms of Gene Expression in the Pathologies of Hyperhomocysteinemia"

_ijms, 2019, doi:10.3390/ijms20133140_

Round 1

Reviewer 1 Report

This review addresses an interesting and relevant topic pertaining to the relationship between homocysteine elevation and derangement of epigenetic mechanisms in disease. It is comprehensive and well written. However, the manuscript should be revised in light of the following points:

- Lines 82 – 84: the authors cite only one paper, whereas other published papers do highlight and present evidence for disturbed DNA methylation in hyperhomocysteinemia (HHcy). Furthermore, in the paper cited by the authors, Homocysteine (Hcy) elevation seems to have an effect on global protein methylation, likely via inhibition of protein methyltransferases. Therefore, the hypothesis that cellular hypomethylation underlies, at least in part, the pathophysiology associated HHcy, should not be dismissed. The authors should reformulate this statement, as well as other statements pertaining to the hypomethylation hypothesis.

- In light of the disappointing Hcy-lowering trials for prevention of cardiovascular events, this topic should merit a brief paragraph in the manuscript. Is it possible that epigenetic derangement initiated by Hcy elevation cannot be reverted by dietary interventions aimed at lowering Hcy? The manuscript would benefit from this discussion in a dedicated small section.

- The tables are overlong and could possibly be provided as supplementary information, rather than included in the article.

Author Response

Response to Reviewer 1 Comments

Point 1: Lines 82 – 84: the authors cite only one paper, whereas other published papers do highlight and present evidence for disturbed DNA methylation in hyperhomocysteinemia (HHcy). Furthermore, in the paper cited by the authors, Homocysteine (Hcy) elevation seems to have an effect on global protein methylation, likely via inhibition of protein methyltransferases. Therefore, the hypothesis that cellular hypomethylation underlies, at least in part, the pathophysiology associated HHcy, should not be dismissed. The authors should reformulate this statement, as well as other statements pertaining to the hypomethylation hypothesis.

Response 1: It was not our intention to dismiss the connection between HHcy and hypomethylation. In our review we present results of studies that show that HHcy does not result in general hypomethylation but rather the effect is gene- and organ-specific. Accordingly, on line 84 additional ref is added to support the statement that reduced SAM/SAH ratio does not always lead to DNA hypomethylation. Other references showing either hypo- or hypermethylation of either DNA or proteins are listed in tables 1, 2, and 3 as well as in sections 3.2. HHcy & SAH/SAM levels and 6.3. Histone modifications and HHcy. In the Introduction we point out that generalization should be avoided because the picture is much more complicated.

Point 2: In light of the disappointing Hcy-lowering trials for prevention of cardiovascular events, this topic should merit a brief paragraph in the manuscript. Is it possible that epigenetic derangement initiated by Hcy elevation cannot be reverted by dietary interventions aimed at lowering Hcy? The manuscript would benefit from this discussion in a dedicated small section.

Response 2: The following sentence has been added to the Summary/prospects section of the manuscript, lines 967-971: “The failure of Hcy lowering therapies to reduce cardiovascular events, may be due to the lack of effect of these treatments on epigenetic changes induced by moderate HHcy in subjects with normal renal function (202, 203). However, in patients with uraemia folate treatment corrects global DNA hypomethylation (116).”

Point 3: The tables are overlong and could possibly be provided as supplementary information, rather than included in the article.

Response 3: Original Tables 1 and 2 are now Supplementary Tables 1 and 2.  Original Tables 3, 4, 5, and 6, now Tables 1, 2, 3, and 4, respectively, provide a useful summary of the effects of HHcy on epigenetic marks, the main topic of our manuscript, and help to orient the reader in the multitude of effects in a variety of model systems.

Reviewer 2 Report

The review is written by an expert in the field, and gives a comprehensive overview of relevant literature.

Authors mention in the abstract  that epigenetic dysregulation of gene expression, mediated by changes in DNA methylation and histone N-homocysteinylation is a pathogenic consequence of HHcy in many human diseases. These findings provide new insights into the mechanisms of human disease induced by Hcy and its metabolites, and suggest therapeutic targets for the prevention and/or treatment. 

However, in the summary/prospects there is very litte info given on therapeutic targets, and nothing on how  authors envisage prevention and/or treatment. That would be interesting to read.

There are a zillion mechanisms and effects on the living organism due to HCY involvement, which is  

described throughout the paper, I would like to see ideas on prevention/treatment options.

There are only a few minor issues, I am not sure but somehow I do not see references for some lines 195 to 199, such as I cite 

DNMT1, essential for maintenance of DNA methylation following DNA replication in cells, is strongly inhibited by SAH, generated during DNA methylation. ref?

' On the other hand, AHCY binds to DNMT1 during DNA replication and enhances DNMT1 activity in vitro'. ref?

'Overexpression of AHCY in mammalian cells leads to hypermethylation of the genome, whereas the inhibition of AHCY has an opposite effect. ' ref?

Accordingly, 

'It has been suggested that alteration of AHCY level affects global DNA methylation and gene expression. '  ref 

Maybe I missed the ref order, and references are listed in other places, but I could not find it.

Some papers dealing with these issues are Baric et al 2004, PNAS, and Motzek et al 2016 Plos One.

In terms of liver cancer, increased Hcy, AHCY deficiency: Stender et al 2015, doi: 10.1016/j.ymgme.2015.10.009

Line 933

In terms of H19, there could also be a reference added, for example I did not see Castello et al. 2019, Mol Cell (https://doi.org/10.1016/j.molcel.2016.06.029).  

I found some typo:

Line 934: 'hydrolase', should be 'hydrolyse...'

Author Response

Response to Reviewer 2 Comments

Point 1: However, in the summary/prospects there is very little info given on therapeutic targets, and nothing on how authors envisage prevention and/or treatment. That would be interesting to read. There are a zillion mechanisms and effects on the living organism due to HCY involvement, which is described throughout the paper, I would like to see ideas on prevention/treatment options.

Response 1: Regarding prevention/treatment options, the following has been included in the Summary/prospects section of the manuscript:Accumulating evidence suggests that targeting HTL, which is involved in the pathology of HHcy (26, 200), might be an effective strategy for disease prevention. One possible therapeutic approach is to pharmacologically inhibit MARS using Hcy analogues, which reduce HTL synthesis at the MARS active site, which in turn attenuates protein N-homocysteinylation (8). Another approach would be to up-regulate HTL-hydrolyzing enzymes, such as PON1, BLMH, or BPHL, which also would reduce HTL levels and N-Hcy-protein accumulation (201, 202).”

Point 2: There are only a few minor issues, I am not sure but somehow I do not see references for some lines 195 to 199, such as I cite DNMT1, essential for maintenance of DNA methylation following DNA replication in cells, is strongly inhibited by SAH, generated during DNA methylation. ref?

' On the other hand, AHCY binds to DNMT1 during DNA replication and enhances DNMT1 activity in vitro'. ref?

'Overexpression of AHCY in mammalian cells leads to hypermethylation of the genome, whereas the inhibition of AHCY has an opposite effect. ' ref?

Accordingly,

'It has been suggested that alteration of AHCY level affects global DNA methylation and gene expression. '  ref

Maybe I missed the ref order, and references are listed in other places, but I could not find it.

Some papers dealing with these issues are Baric et al 2004, PNAS, and Motzek et al 2016 Plos One.

In terms of liver cancer, increased Hcy, AHCY deficiency: Stender et al 2015, doi: 10.1016/j.ymgme.2015.10.009

Line 933

In terms of H19, there could also be a reference added, for example I did not see Castello et al. 2019, Mol Cell (https://doi.org/10.1016/j.molcel.2016.06.029). 

Response 2: Refs for the statement “DNMT1, essential for maintenance of DNA methylation following DNA replication in cells, is strongly inhibited by SAH, generated during DNA methylation” has been added: The specificity of S-adenosylmethionine derivatives in methyl transfer reactions by Zappia et al. JBC 1969 and SAR around (L)-S-adenosyl-L-homocysteine, an inhibitor of human DNA methyltransferase (DNMT) enzymes by Saavedra et al. Bioorg Med Chem Lett 2009.

The reference for the following statements:

' On the other hand, AHCY binds to DNMT1 during DNA replication and enhances DNMT1 activity in vitro'.

'Overexpression of AHCY in mammalian cells leads to hypermethylation of the genome, whereas the inhibition of AHCY has an opposite effect. '

'It has been suggested that alteration of AHCY level affects global DNA methylation and gene expression. '

is: S-adenosylhomocysteine Hydrolase Participates in DNA Methylation Inheritance Ponnaluri 2017.

In terms of H19, there could also be a reference added, for example I did not see Castello et al. 2019, Mol Cell (https://doi.org/10.1016/j.molcel.2016.06.029). 

These refs cannot be included because they are not relevant to the subject of our review: 10.1016/j.molcel.2016.06.029 is Comprehensive Identification of RNA-Binding Domains in Human Cells by Castello et al. 2016. It is on RNA binding domains in human cells, but not H19. Castello et al. Mol Cell 2019 is on a similar subject, not relevant to our manuscript.

Point 3: I found some typo:

Line 934: 'hydrolase', should be 'hydrolyse...'
.

Response 3: The typo has been corrected.